



# The first pan-Alpine surface-gravity database, a modern compilation that crosses frontiers

Pavol Zahorec[1], Juraj Papčo[2], Roman Pašteka[3], Miroslav Bielik[1,3], Sylvain Bonvalot[4], Carla Braitenberg[5], Jörg Ebbing[6], Gerald Gabriel[7], Andrej Gosar[8], Adam Grand[3], Hans-Jürgen Götze[6], György Hetényi[9], Nils Holzrichter[6], Edi Kissling[10], Urs Marti[11], Bruno Meurers[12], Jan Mrlina[13], Ema Nogová[1,3], Alberto Pastorutti[5], Matteo Scarponi[9], Josef Sebera[6], Lucia Seoane[4], Peter Skiba[7], Eszter Szűcs[14], Matej Varga[15]

1 Earth Science Institute, Slovak Academy of Sciences, Dúbravská cesta 9, 840 05 Bratislava, Slovakia
2 Department of Theoretical Geodesy, Faculty of Civil Engineering, Slovak University of Technology in Bratislava, Radlinskeho 11, 810 05 Bratislava, Slovakia
3 Department of Applied and Environmental Geophysics, Faculty of Natural Sciences, Comenius University in Bratislava, Mlynska dolina, Ilkovicova 6, 842 48 Bratislava, Slovakia
4 Bureau Gravimétrique International, Toulouse, France
  GET, University of Toulouse, CNRS, IRD, UT3, CNES, Toulouse, France
5 Department of Mathematics and Geosciences, University of Trieste, Via Edoardo Weiss 1, 34128 Trieste, Italy
6 Institute of Geosciences, Christian-Albrechts-University Kiel, Otto-Hahn-Platz 1, 24118 Kiel, Germany
7 Leibniz Institute for Applied Geophysics, Stilleweg 2, 30655 Hannover, Germany
8 Slovenian Environmental Agency, Seismology and Geology Office, Vojkova 1b, 1000 Ljubljana, Slovenia and University of Ljubljana, Faculty of Natural Sciences and
  Engineering, Aškerčeva 12, 1000 Ljubljana, Slovenia
9 Institute of Earth Sciences, University of Lausanne, UNIL-Mouline Géopolis, 1015 Lausanne, Switzerland
10 Department of Earth Sciences, Federal Institute of Technology (ETH), Sonneggstrasse 5, 8092 Zürich, Switzerland
11 Federal Office of Topography swisstopo, Wabern, Switzerland
12 Department of Meteorology and Geophysics, University of Vienna, 1090 Vienna, Althanstraße 14, UZA 2, Austria
13 Institute of Geophysics, Czech Academy of Sciences, Boční II/1401, 141 31 Prague, Czech Republic
14 Geodetic and Geophysical Institute, RCAES, Hungarian Academy of Science, Csatkai street 6-8., 9400 Sopron, Hungary
15 Faculty of Geodesy, University of Zagreb, Kačićeva 26, 10000 Zagreb, Croatia

*Correspondence to*: Hans-Jürgen Götze (hajo.goetze@ifg.uni-kiel.de)

**Abstract.** The AlpArray Gravity Research Group (AAGRG), as part of the European AlpArray program, focuses on the compilation of a homogeneous surface-based gravity dataset across the Alpine area. From this data set, Bouguer- and Free Air anomalies are calculated and presented here. In 2016/17 ten European countries in the Alpine realm have agreed to contribute with gravity data for a new compilation of the Alpine gravity field in an area from 2° to 23° East and from 41° to 51° North. This compilation relies on existing national gravity databases and, for the Ligurian and the Adriatic seas, on ship-borne data of the Bureau Gravimétrique International. Furthermore, for the Ivrea zone in the Western Alps, recently acquired data were added to the database. This first pan-Alpine gravity data map is homogeneous regarding input data sets, applied methods and all corrections as well as reference frames.





Here, the AAGRG presents the data set of the recalculated gravity fields on a 4 km × 4 km grid for public release, 2 km × 2 km for special request. The final products also include calculated values for mass/bathymetry corrections of the measured gravity at each grid point, as well as height. This allows users to use later customized densities for their own calculations of mass corrections. Correction densities used are 2670 kg m$^{-3}$ for landmasses, 1030 kg m$^{-3}$ for water masses above and -1640 kg m$^{-3}$ below the ellipsoid. The correction radius was set to the Hayford zone O$_2$ (167 km). The new Bouguer anomaly is station

completed (CBA) and compiled according to the most modern criteria and reference frames (both positioning and gravity), including atmospheric corrections. Special emphasis was put on the gravity effect of the numerous lakes in the study area, which can have an effect of up to 5 mGal for gravity stations located at shorelines with steep slopes, e.g., for the rather deep reservoirs in the Alps. The results of an error statistic based on cross validations and/or "interpolations residuals" is provided for the entire database. As an example, the interpolation residuals of the Austrian data set range between about -8 and +8 mGal,

the cross-validation residuals between -14 mGal and +10 mGal; standard deviations are well below 1 mGal. The accuracy of the newly compiled gravity database is close to ± 5 mGal for most areas.

A first interpretation of the new map shows that the resolution of the gravity anomalies is suited for applications ranging from intra-crustal to crustal scale modelling to interdisciplinary studies on the regional and continental scales as well as applications as joint inversion with other datasets.

The data will be published with the DOI https://doi.org/10.5880/fidgeo.2020.045 (Zahorec et al., 2020) when the final paper is accepted. In the meantime, the data is accessible via this temporary review link: https://dataservices.gfz-potsdam.de/panmetaworks/review/fdc35a9f6551b01b6152ee1af7b91a5a0c3de5341d067644522c192ad7f25e7f/

**Abbreviations**

| | |
|---|---|
| AAGRG | AlpArray Gravity Research Group |
| BC | Bathymetric correction |
| BEV | Federal Office of Metrology and Surveying, Vienna, Austria |
| BGF | Banque Gravimétrique de la France |
| BGI | Bureau Gravimetrique International |
| BRGM | Bureau de Recherches Géologiques et Minières |
| CAGL | Central Apennine gravity low |
| CBA | Complete Bouguer anomaly |
| CGF65 | Carte Gravimétrique de la France 1965 |
| CGG | Compagnie Générale de Géophysique |
| CNEXO | Centre National pour l'Exploitation des Océans |
| CV | Cross validation |
| DEM | Digital elevation model |
| DEM25 | Digital elevation model (25 meter resolution, Germany) |
| DGL | Dinaric gravity low |
| DHHN | German main levelling network |
| DTM | Digital Terrain Model |
| DRE | Distant relief effect |
| EGM2008 | Earth Gravitational Model of 2008 |



|  |  |  |
|---|---|---|
|  | EIGEN (6C4) | European Improved Gravity model of the Earth by New techniques (6C4) |
|  | EMODnet | European Marine Observation and Data Network |
| 80 | ETRS89 | European Terrestrial Reference System 1989 |
|  | EOV | Hungarian geodetic coordinates in national map projection |
|  | EVRS | European Vertical Reference System of 2020 |
|  | FA | Free air anomaly |
|  | GEBCO | General Bathymetric Chart of the Oceans |
| 85 | GGM | Global Gravitational Model |
|  | GIE | Geophysical Indirect Effect |
|  | GIS | Geographic information system |
|  | GNSS | Global navigation satellite system |
|  | GPS | Global positioning system |
| 90 | GRAVI-CH | Gravity database of Switzerland |
|  | GRS80 | Geodetic Reference System from 1980 |
|  | HVRS1971 | Croatian Height Reference System from 1971 |
|  | IAG | International Association of Geodesy |
|  | IFREMER | Institut Français de Recherche pour l'Exploitation de la Mer |
| 95 | IGF | International Gravity Formula |
|  | IGFS | International Gravity Field Service |
|  | IGH | Ivrea gravity high |
|  | IGN | Institut de l'Information Géographique et Forestière |
|  | IGSN71 | International gravity standardization net of 1971 |
| 100 | IUGG67 | International Union of Geophysics and Geodesy, 1967 congress |
|  | LCC | Lambert Conformal Conic (projection) |
|  | LiDAR | Light Detection and Ranging |
|  | LN02 | Height system of Switzerland |
|  | MC | Mass correction |
| 105 | MERIT DEM | Multi error removed improved terrain DEM |
|  | MGH | Hungarian gravity network |
|  | MGHi | Mediterranean gravity high |
|  | NAGL | Northern Apennine gravity low |
|  | NTE | Near terrain effect |
| 110 | OGS | National Institute of Oceanography and Experimental Geophysics |
|  | OMV AG | Österreichische Mineralölverwaltung AG |
|  | PBGH | Pannonian Basin gravity high |
|  | RCGF09 | Gravimetric Network and Map of France 2009 |
|  | RGF83 | Réseau Gravimétrique Français |
| 115 | RMS | Root mean square |
|  | RTM | Residual Terrain Modelling |
|  | SAPOS | Satellite Positioning Service (German Surveying and Mapping Agency) |
|  | SDB | Satellite Derived Bathymetry |
|  | SGr-57, 67, 95 | Czech and Slovak National Gravimetric System of 1957, 1967 and 1995 |
| 120 | SKPOS | Slovak real-time positioning service |
|  | SHOM | Service hydrographique et océanographique de la Marine |
|  | SI | Système international d'unités (International unit system) |
|  | SRTM | Shuttle Radar Topography Mission |
|  | TC | Terrain correction |
| 125 | TM | Transverse Mercator (projection) |
|  | UTM | Universal Transverse Mercator (projection) |
|  | VFGL | Venetian-Friuli Plain gravity low |





| WCGL | Western Carpathian gravity low |
| WGS84 | World Geodetic System 1984 |


# 1 Introduction

## 1.1 History

There is a long history of geological and geophysical research on the Alpine orogen, the results of which point to two main

groups of complexity. The first is the temporal evolution of the mountain belt, with plates, terrains and units of different size

and level of deformation, mostly investigated from the geological record (e.g., Handy et al., 2010). This inheritance directly

influences the second level of complexity, which is structural and characterizes every level of the lithosphere from sedimentary

basins to orogenic roots and also the upper mantle. The level of along-strike variability of the Alps exceeds what is known in

other mountain belts such as the Andes and the Himalaya (Oncken et al., 2006; Hetényi et al., 2016), and explains why some

of the orogenic processes operating in the Alps are still debated.

Structural complexity at depth, and thus advance in our understanding of orogeny, can be resolved by high-resolution 3D

geophysical imaging. These are among the primary goals of the AlpArray program, and its main seismological imaging tool,

the AlpArray Seismic Network. This modern array has used over 628 sites for more than 39 months across the greater Alpine

area such that no point on land was farther than 30 km from a broadband seismometer (Hetényi et al., 2018). While seismic

imaging of the entire Alps in 3D became reality following decades of active- and passive-source projects, imaging efforts in

gravity reached 3D earlier, thanks to the availability of national data sets of the Alpine neighbouring countries with partly high

resolution and 3D modelling approaches among others (Ehrismann et al., 1976; Götze, 1978; Kissling, 1980; Götze and

Lahmeyer, 1988; Götze et al., 1991; Ebbing, 2002; Ebbing et al., 2006; Marson and Klingelé, 1993; Kahle and Klingelé, 1979).

However, these land data sets, for historical reasons, were acquired in national reference systems, and were seldom shared,

preventing high-resolution pan-Alpine gravity studies using homogeneously processed data.


## 1.2 History of gravity mapping in the Alps

Despite the sometimes enormous terrain conditions in the Western and Eastern Alps, the Alpine countries developed efforts

to obtain information on the gravity field of the Alps by means of geodetic and geophysical measuring methods at an early

stage. Most of the expertise developed in this process was incorporated into the compilation of the new digital gravity bases.

Here is a very brief overview of the historical activities of the main actors, in Sect. 2 the national contributions to the pan-

Alpine Bouguer gravity map are listed.



### Austria

Zych (1988) reports on the first gravity measurements in Austria in the course of hydrocarbon exploration as early as 1919, while more intensive, regional and detailed measurements were carried out in the following years with pendulums, torsion balances and gravimeters, concentrating mainly on the Vienna Basin and neighbouring areas. This and other measurements were later included in the gravity map of Austria (Senftl, 1965) by the Federal Office of Metrology and Surveying (BEV), at a scale of 1 : 1 million. BEV, several universities in Austria (Vienna, Leoben) and Germany (Clausthal-Zellerfeld) as well as hydrocarbon industry (OMV AG, Austria) added numerous gravity profiles and areal networks across the Austrian territory since then (see e.g. Meurers, 1992a and b; Steinhauser et al., 1990; Götze et al., 1979). In 2009, Meurers and Ruess published a complete review of the gravity values measured in Austria, "A new Bouguer Gravity Map of Austria" (Meurers and Ruess, 2009) on the basis of 54 000 land gravimetric data. These recompilations already contained most of the numerical approaches that have been implemented in our new Pan-Alpine Gravity Map.

### Switzerland

An early compilation of gravity measurements and a gravity map covering the entire country was published in 1921 based on data acquired since 1900 (Niethammer, 1921). In 2008, the Institute of Geophysics of the University of Lausanne published the gravity map of Bouguer anomalies in Switzerland 1 : 500 000 for the Swiss Geophysical Commission: editors were Olivier et al. (2010) and their compilation based on the work of Klingelé and Olivier (1980). It reflects the culmination of more than 15 years of work and effort on the part of many staff and students at the Geophysical Institutes of the University of Lausanne and the Polytechnic School of Zurich. Between 1994 and 2002, a set of twenty-two 1 : 100 000 scale maps of Bouguer anomalies was published. The anomalies were calculated with the 1967 ellipsoid, with a density of 2670 kg m$^{-3}$, and corrected for relief up to a distance of 167 km around each station. These maps were elaborated from 29 900 measured stations selected from the gravity database GRAVI-CH over a territory of about 56 000 km$^2$. In total, approx. 85 gravimetric campaigns were carried out between 1986 and 2000. The Swiss experience with the Bouguer gravity compilation was also exemplary for the creation of a common gravity database in the entire Alpine region.

### France

A detailed and systematic gravimetric coverage of the French territory was conducted in the frame of the Carte Gravimétrique de la France 1965 (CGF65). The establishment of a reference network of 2000 base stations originally linked to international absolute stations (Potsdam system) and the gravity surveys carried out between 1945 and 1975 using North American, LaCoste & Romberg and Worden meters for mapping, mineral and oil prospecting or for academic purposes provided the first gravity infrastructure at national scale. Despite incomplete coverage, it was published in 1975 in the form of a map on a scale of 1:1 000 000 (North and South sheets). The primary reference network was later updated as the Réseau Gravimétrique Français (RGF83) with additional absolute gravity measurements and link to the IGSN71 international network. The digital recording





of available terrestrial gravity data acquired by several organizations (Bureau de Recherches Géologiques et Minières, BRGM; Institut de l'Information Géographique et Forestière, IGN; Oil and mining companies; Universities and research institutes), was started in 1977. In 1990, BRGM founded the "Banque Gravimétrique de la France, BGF" to manage and update the stations on the French gravity map. The BRGM database is also periodically replicated to the "International Gravimetric Bureau, BGI" for data distribution and contribution to the global gravity mapping.

**Italy**

The history of gravity measurements worldwide and especially in Italy began with the free fall experiments of Galileo Galilei (1564 - 1642). In his honorary capacity we still use Gal or mGal ($10^{-5}$ m s$^{-2}$) until today (Marson, 2012). The eighties and nineties of the twentieth century were characterized by the development of an own absolute gravity meter (Istituto di Metrologia G. Colonnetti), on- and offshore measurements (Gulf of Naples and 2000 km gravity profiles in the Mediterranean Sea) in connection with European geodesy projects (Morelli and Sansò, 1994).

In 1975 the late Italian Geodetic Commission decided on the compilation of a new Bouguer anomaly map of Italy based on up to date correction standards and homogeneous methodology. This map was published in 1991 by the National Research Council (C.N.R.-P.F.G., 1991) as part of the Structural Model of Italy at a scale of 1 : 500 000. The gravity values were referred to IGSN-71 (Morelli et al., 1972), and density for the terrain correction was set to 2400 kg m$^{-3}$, and the main data contribution was from the Italian National Oil Company (ENI-AGIP).

In 1989 the Geological Survey of Italy together with ENI - AGIP published a new gravity map of Italy scaled 1 : 1 000 000 using the dataset collected for the 1 : 500 000 CNR gravity map. In 1975 the later Italian Geodetic Commission set up a new Bouguer anomaly map of Italy based on up-to-date correction standards and homogeneous methodology. In the 1990's the Geological Survey undertook an extensive land gravity cartography program that should cover the whole national territory at the scale of 1 : 50 000. The presently available gravity map from the National Environmental Agency (APAT) – Department of Terrain defence - National Geological Survey is a map published at the scale of 1 : 1 250 000 published in 2005 (APAT, 2005; Ferri et al., 2005), which used a terrain correction density of 2670 kg m$^{-3}$ and the Hayford radius of 166.736 km. Data were collected from different sources, as ENI, OGS, and the U.S. Defence Mapping Agency, academic organisations and the former Italian Geological Survey. Station density in the Alps for this map is about 0.1 to 0.2 stations per 1 km², and it increases to 1.5 stations per 1 km² in the basins. The Bouguer anomaly has been corrected for topography onshore, whereas for offshore a free air anomaly map was published.

**Slovenia**

The first map of Bouguer anomalies which comprises the whole Slovenian territory was compiled in 1967 (Čibej, 1967; Ravnik et al., 1995). It was based on data measurements with a Worden gravity meter (no. 117) in the framework of various gravity surveys conducted over the period 1952-1965 by the Geological Survey of Slovenia (Stopar, 2016). Later in the frame of the W-E Europe Gravity Project led by Getech from Leeds University a new dataset was prepared in 1990's which comprises 416





gravity points giving an average density of 0.02 gravity stations per 1 km². Gravity data in Slovenia reflect a complex structural setting in the transition area between the Alps, Carpathians, Dinarides and Pannonian basin. Large variations in the crustal thickness (Gosar, 2016) and the depth of sedimentary basins in the transition from the Alps-Dinarides to the Pannonian basin in Slovenia are clearly reflected in Bouguer anomalies.

**Germany**

With the start of the "Deutsche Reichsaufnahme" in 1934, an important development phase began also for gravity in Germany. Gravimetric maps were produced by the "Amt für Bodenforschung" and supplemented mainly for the Alpine foreland. After 1945, the "Amt für Bodenforschung" coordinated first efforts to complement this database in West Germany. Gerke (1957) published the gravity map of West Germany (cited after Closs, 2008). The Bouguer gravity map 1 : 500 000 of the Federal Republic of Germany was produced by S. Plaumann in 1995 (e.g. sheet South - now referred to IGSN71) on the basis of

measurements by the "Geophysical Survey of the Federal Republic of Germany, the Lower Saxony State Office for Soil Research and oil companies". After corrections of the gravity meter drift and terrain, they were reduced to sea level with a density of 2670 kg m$^{-3}$ and referred to IGSN71. Based on more than 275 000 data points, current reference systems, improved terrain models, and the computing power available today, Skiba et al. (2010) compiled the current Bouguer gravity map and oriented themselves to the current international standards of neighbouring countries.

**Slovak Republic**

  A thorough overview of the practical and methodological developments of gravimetry in the Slovak Republic can be found in "Understanding the Bouguer Anomaly - A Gravimetry Puzzle" (Pašteka et al., 2017). The territory of the Slovak Republic (except the inaccessible areas of the Tatra Mountains) is covered by regional gravity measurements in the scale 1 : 25 000 with station spacing from 3 to 6 stations per 1 km². The measurements were realized during a long period from the 1950s up to the

1990s. The project goal was to create a high definition gravity map for mineral exploration and basic geologic interpretations. Various types of gravity meters were used during the data acquisition time period (GAK PT, Worden, Canadian CG-2, Scintrex CG-3M). Different approaches to complete Bouguer anomaly (CBA) calculation were used, including different normal field formulas, different equations for "Bouguer" correction and atmospheric correction, as well as various methods of the terrain correction estimation. A complete recalculation of the entire database was performed in the frame of the earlier project Atlas

of geophysical map and lines (Grand et al., 2001). Several hundreds of points with errors in their heights or positions were identified - these points had been removed from the final Bouguer anomaly evaluation.

**The AlpArray Gravity Research Group**

  With respect to the national expertise and databases available in the Alpine countries, the formation of an international Research Group was decided within the framework of activities in the European AlpArray program and established at an EGU

Splinter Meeting in 2017. In the subsequent workshops in Bratislava (Slovakia) in 2018, and two further technical meetings

of the group (again in Bratislava in 2018 and in Sopron (Hungary) in 2019), the organisational, scientific, and numerical requirements for the compilation of the new pan-Alpine digital gravity database were established which consists of Bouguer- and Free-Air anomalies and values of mass correction. Although most of the national group members were extensively involved in the processing of data, we would like to remind gratefully that by far the most intensive part of the processing was

done by the group members from Bratislava and Banská Bystrica (Slovakia).

In the following, we present our effort, omitting historical obstacles, in compiling and merging all available land and sea gravity data in the greater Alpine area, a total of more than 1 million on- and offshore data points. We commit to the exact same data processing procedures, so that even proprietary point-wise data can be included at the project's initial stage and represented in the final data grids.

**1.3 Layout of the publication**

The following section document in detail our procedures, from raw data to final high-resolution gravity maps. The referencing and quality assessment of various gravity databases and digital Earth surface models are discussed in Sect. 2. The equations and their implementations to obtain various gravity anomaly products are described in Sect. 3. Reprocessing of original raw data and of the related corrections were carried out (Sect. 4). Sect. 5 presents the new, homogenized Bouguer gravity map for

the Alps. In Sect. 5.3 we describe the attached Bouguer map together with an accompanying description and interpretation of the gravity anomalies in the Alps and their surroundings. Notes on the uncertainty of the compilation are given in Sect. 6. We conclude on the listing and availability of the new gravity data (Sect. 7), which we share publicly as a contribution to further gravity studies in the region at different scales.

**2 Assessment of Database**

Here we describe the initial situation for the assessment and application of existing data, available publications, data density and quality description country by country.

> **Note: Different from the SI units we will use here the unit mGal for gravity, which is still frequently used in gravimetry; 1 mGal = $10^{-5}$ m s$^{-2}$.**

The following partner and AAGRG member countries have contributed to the compilation of the new Pan-alpine gravity maps:

**Austria**

In the early beginning, gravity stations in Austria were mainly arranged along levelling lines. The first areal network, which was surveyed by OMV, focused on the Alpine Foreland, the Vienna basin and parts of the Flysch and Calcareous zone of the Eastern Alps (Zych, 1988). Additional gravity profiles were established across the central part of the Eastern Alps (Ehrismann et al., 1969, 1973, 1976; Götze et al., 1978) 50 years ago. The vertical coordinates of all stations so far were determined by

precise levelling, while horizontal coordinates were based on topographic map digitization providing an accuracy estimate of



±25 m. The first area station design with stations even on high mountain flanks and peaks started during the late 1970ies (Götze et al., 1979; Schmidt, 1985; Meurers et al., 1987; Posch and Walach, 1990; Walach, 1990; Walach and Winter, 1994). Most of the new stations were established at benchmarks of the national cadastre with maximum coordinate errors of a few 10 cm in height and even better accuracy in horizontal position, even on high mountains. However, in large areas, particularly along

the Alpine crest, station coverage was sparse. Since 1982, GPS techniques and helicopter transportation in otherwise unaccessible mountainous regions made also these areas accessible while meeting modern accuracy requirements. Presently the Austrian gravity database contains about 54 000 stations with an average station interval of less than 3 km even in the high mountains and average station density of 1 station per 9 km$^2$ or higher. In the early gravity campaigns Askania and Worden gravimeters were used, since 1970 only LaCoste & Romberg or Scintrex gravimeters. Depending on the data provider and

acquisition date, data referred to different datum and exhibit different accuracy. In addition, industrial data (OMV) was tied to an own gravity base which had a slightly different scale due to limited calibration accuracy. For the most recent gravity map of Austria (Meurers and Ruess, 2009) all data were homogenized regarding height and gravity datum based on ties to the Austrian absolute gravity network (Ruess, 2002; Meurers and Ruess, 2007). Gross coordinate errors were detected by comparing station heights with interpolations of a high-resolution digital terrain model with 50 m spacing. Erroneous

coordinates were corrected by using modern topographic and orthophoto maps and by utilizing the digital cadastre (Meurers and Ruess, 2007). Based on modern methods of terrain correction procedures, digital terrain models and a new geoid model (Pail et al., 2008), the Bouguer anomaly of Austria was determined using for the first time ellipsoidal heights (Meurers and Ruess, 2009). The exact transformation from local Gauß-Krüger coordinates and orthometric heights into ETRS89 UTM and WGS84 geographical coordinates was done by applying a stepwise procedure recommended by the national surveying office

(BEV, www.bev.gv.at).

**Croatia**

The Croatian national gravity database consists of approximately 16 500 Free-Air anomaly values covering the entire continental area. Data in the database were mainly collected from 1945 to 1990 across the territory of the former Socialist Federal Republic of Yugoslavia (SFRY). The data are almost equally distributed across the wider territory of Croatia, also

including some points in Bosnia and Herzegovina and Slovenia. The average point density is 1 point per 18 km$^2$; in continental part of Croatia data density is 1 point per 8 km$^2$, whereas in mountainous areas and on islands density is much lower (1 station per 30 km$^2$). Each point has appended geodetic coordinates referring to GRS80 ellipsoid, whereas heights are normal-orthometric referring to the national height reference system Croatian Height Reference System (HVRS1971). Gravity values refer to the International Gravity System Network of 1971 (IGSN71). Metadata about the accuracy of gravity values, position,

and heights does not exist. Since its creation the database passed through several phases of checking, cleaning, debiasing and filtering. It was used in geophysics for creating Bouguer anomaly maps (Bilibajkić, 1979) in the past. Most recently, it found specific usage in national geoid model determination (Bašić, 2009; Varga, 2018). For the purposes of AAGRG project all available points were included in gridding of the model of Bouguer anomalies.



**Czech and Slovak Republic**

Equally for the Czech and the Slovak Republic, most regional gravity surveys were conducted in the 1950s till 1990s. Prevailing sampling interval was about 500 m, or 5 stations per 1 km$^2$, during the so-called "mapping 1 : 25 000" scale. This mapping covered about 75% of the Czech Republic and 100% of the Slovakia territory, while the rest was previously covered by mapping 1 : 200 000 scale with about 1 station per 4 km$^2$. Principal targets of the surveys were mineral exploration for uranium, tin and other minerals, oil and gas, hydrological and environmental investigations, as well as basic geological

research. The database was reduced to a 2 × 2 km coverage and contains now 13 955 points for the Czech Republic and 21 108 points from the Slovak Republic. Positions of the stations were digitized from the "Military Topographical Maps" at the scale 1 : 25 000 in a Gauss-Krüger projection coordinate system. Accuracy in position of these points is in the range of 10 - 50 m. Heights of the gravity points were determined in Balt vertical reference system by geodetic levelling connected to the points of the National levelling network. Vertical accuracy ranges from 5 cm in the lowlands to 50 cm in the mountains. Gravity

values were tied to the "National Gravimetric System SGr-57, 67" which is connected to the old Potsdam system. Consequently, they were transformed to the recent absolute gravimetric system SGr-95. Accuracy of the gravity values is up to 100 μGal.

Further parameters of this exemplary new compilation are the use of the Somigliana-Pizzetti formula for normal gravity, spherical calculation of the topography effect (density 2670 kg m$^{-3}$), Free air correction term and atmospheric correction. In

addition to the mentioned standard steps of the CBA calculation, effects of the distant topography, bathymetry and ice sheet effects were calculated for the entire database. The expertise gained was fully available for the compilation of the alpine gravity map.

One of the most important steps of this process is the precise evaluation of the terrain corrections. For selected areas of Slovakia gravity maps were compiled and purpose derived gravity maps and density models were constructed along selected regional

gravimetric profiles across the territory of the Western Carpathians. The first map in Czech Republic was made accessible to the public in April, 2009, last updated in April, 2013 and turned into a world-wide-web format implemented in 2014.

**France**

Since the early '90s, gravity densifications have been realized using Scintrex gravity meters (CG3, CG5 and currently CG6) and accurate GPS positioning, mainly as part of major scientific projects such as GéoFrance3D ("Millennium Project"). A new

gravity database based on both recalculated corrections with a density of 2670 kg m$^{-3}$ and on the IGSN71 system using data from the BGF and other sources (Grandjean et al., 1998) was established. A new gravity map of France, including terrain corrections uniformly computed up to 166.7 km, was released by BRGM (Martelet et al., 2009) in the frame of the RCGF09 action (Gravimetric Network and Map of France 2009), which also led to the joint creation of a new gravimetric network by IGN. Since 2006, hybrid relative (Scintrex) and absolute (Micro-g A10) gravity surveys have been carried out by IGN for

defining a 1st order precise gravity reference network (RMS 25 μGal) of over 1200 stations. Nowadays, the complete gravity





coverage of the French territory contains approximately 370 000 points. All this gravity information is currently used to refine the computation of the national geoid, of the gravity anomalies and of the height conversion grids.

The gravity datasets over France and the surrounding marine areas are provided from the BGI global gravity databases (http://bgi.obs-mip.fr/). Terrestrial data are mostly derived from the gravity surveys carried out and compiled by BRGM. They

also include 2272 gravity data points in the Alps provided by IGN and other contributions from by Guglielmetti et al. (2013) and research laboratories (Paris, Toulouse, Montpellier, Strasbourg, Clermont-Ferrand, Grenoble and Nice). Finally, the dataset has been sampled with 1 point per 4 km$^2$ giving a total amount of 22 593 free air gravity values over the concerned French territory.

Offshore gravity data included in the AlpArray solution are provided by the GEOMED2 project (Lequentrec-Lalancette et al.,

2016; Barzaghi et al., 2018). This project was recently conducted in the frame of the International Association of Geodesy (IAG) by the International Gravity Field Service (IGFS) and BGI, aimed at providing high resolution geoid and gravity grids and maps of the whole Mediterranean Sea. The compilation, validation and adjustment of the above-mentioned French and Italian marine gravity surveys was done by SHOM and BGI considering the usual protocols applied at SHOM (Service Hydrographique et Océanographique de la Marine) for the qualification of marine gravity data. The final GEOMED2 product

led to the realization of a 1' × 1' free air gravity grid for the whole Mediterranean Sea given in the IGSN71 reference system with an estimated accuracy of 3.6 mGal deduced from the internal and external Cross Over Analysis. Details on the gravity data acquisition and compilation can be found in Lequentrec-Lalancette et al. (2016).

**Offshore data of BGI**

Offshore gravity measurements in the study area were collected from shipborne surveys performed since the '60s in the Gulf

of Lyon and Ligurian sea by the French IFREMER, CNEXO, SHOM and CGG. In addition, this area is also covered by the extensive gravity surveys carried out between 1961 and 1972 by the Italian Experimental Geophysical Observatory over the whole Mediterranean Sea and known as the "Morelli dataset" (Allan and Morelli, 1972; Allan et al., 1962). These surveys were conducted with different generations of sea gravity meters (LaCoste & Romberg, Graf-Askania, Bodensee) mounted on a gyro-stabilized platform. Corresponding gravity data and reports are archived by IFREMER and SHOM and transmitted to

the BGI.

**Germany**

The German data used in the AlpArray project originate from three main datasets that were acquired between ca. 1930 and 2010. The AlpArray area is covered by 36 442 gravity stations. As only few historical measurements were carried out in the frame of dense local surveys, the mean point spacing is in the order of 2 to 3 km. Regional gravity measurements were either

conducted at public geodetic reference points, for which precise coordinates were available, or at prominent points that could be easily identified in maps and for which coordinates were digitized. Hence, the precision of the coordinates can vary between some centimetres and some few tens of meters. The heights of the German gravity stations are referred to the reference system

DHHN (German main levelling network), in the version valid at the time of the measurement. This may result in deviations to the current reference system DHHN2016 in the order of some centimetres. During the reprocessing in 2010, station heights

were checked for plausibility by a comparison with heights taken from the DEM25 (the best German DEM at that time). As large deviations can also result from imprecise horizontal coordinates of the stations, such stations were additionally evaluated with respect to their location by means of GIS techniques and, if necessary, by an additional comparison with georeferenced digital topographical maps and orthophotos. For 95% of the stations covering the entire German territory the differences in height are less than 2 meters. Gravity stations that exhibit differences of more than 5 m to DEM25 were not considered in the

data contribution for the compilation of the new AlpArray Bouguer gravity map.

The current Bouguer anomaly map for Germany (Leibniz-Institut für Angewandte Geophysik, 2010; Skiba, 2011), based on more than 275 000 data points, refers to the IGSN71 and a density of 2670 kg m$^{-3}$. Absolute gravity values that were acquired in the old Potsdam gravity system were transferred to the IGSN71. The accuracy of the absolute gravity is estimated to be better than 100 µGal.

*For the AlpArray compilation, gravity data was provided by the Leibniz Institute for Applied Geophysics (including data from the Geophysikalische Reichsaufnahme), Kiel University, and the Geological Survey of Saxony (LfULG).*

**Hungary**

Gravity field investigations and field observations in Hungary were already established by the pioneering work of Baron Loránd (Roland) Eötvös. The Eötvös torsion balance became the world's first geophysical tool for prospecting and revealed

hundreds of hydrocarbon resources. Hungary contributed to the unified Bouguer gravity map with gridded data of 2 km × 2 km given in Gauss-Krüger map projection, the terrain correction was calculated up to a distance of 22.5 km around each station utilizing a uniform reduction density of 2670 kg m$^{-3}$.

The Hungarian gravity database consists of approximately 388 000 data points and covers the whole country with rather heterogeneous point density. Gravity measurements were mainly carried out between 1950 and 2010 with different purposes,

which determines the point distribution. For the oil industry, local exploration grids were established with a few hundred meters grid spacing, on the other hand due to transportation requirements early measurements were arranged along roads. The average point density of 2.8 points per 1 km$^2$ suggests a fair coverage, but it concentrates to areas with low to moderate topography. The database consists of geodetic coordinates given in national map projection (EOV) referred to the IUGG67 ellipsoid, whereas heights are given in Baltic height system. Gravity values are tied to the Hungarian gravity network MGH

(from Hungarian abbreviation), which was established, extended and re-adjusted in several epochs (MGH-50, MGH-80, MGH-2000, MGH-2010 and MGH-2013 networks; Csapó and Völgyesi, 2002; Csapó and Koppán, 2013; Csapó, 2013) to unify gravity values, support regional-scale data processing and connection to the Unified European Gravimetric Network. Metadata on the accuracy of horizontal position, height and gravity data is not provided in the data set. The estimated accuracy of g-values is 0.1 mGal on average. The database was collected and is maintained by the Mining and Geological Survey of Hungary.

Following the requirements for the new pan-Alpine Bouguer model, the high resolution national digital elevation model with





spacing of 30 m × 30 m was used in the computation of the gravitational effect of nearby terrain masses. The DEM was produced by digitizing the isolines of the topographic maps of scale 1 : 10 000.

**Italy**

The Italian data used in the AlpArray project originate from one main dataset, which is industry data handed over by ENI, and several other minor datasets including the Province of Bolzano, newly acquired data in Ivrea-Verbano zone (Scarponi et al., 2020), data acquired in the Province of Bolzano during the INTAGRAF project, and Swiss-topo data. The AlpArray area is covered by 130 905 gravity stations, of which the ENI dataset has 128 479 stations on land and offshore, in the Province of Bolzano there are 1737 stations, and in the Ivrea-Verbano area 689 stations. The data are very dense in the Po-plain, and scarcer in the higher elevations, with a mean point spacing of 705 m. Gravity measurements other than ENI were conducted at cadastral geodetic reference points, for which precise coordinates were available, or were acquired in position and height with parallel GNSS observations. The ENI data points were acquired with either traditional geodetic survey, or the newer points with GNSS. The positions of the Italian gravity stations are referred to the reference system GRS80, with the industry data having been transferred to GRS80 in the frame of a revision of the database, with the heights in normal heights. Geoidal heights were converted to ellipsoidal heights by adding the ITALGEO geoid heights. We have compared the normal heights with different terrain models, with MERIT (Yamazaki et al., 2017) and in the Region Veneto with the local high resolution DEM. The average difference with MERIT of the entire database is 0.3 m, the root mean square difference is 12.63 m. The criterion for using a data point for the final map was a difference with MERIT of less than 50 m. This high height difference is limited to relatively higher elevations, outside the plains, and is probably due rather to the sparse grid-spacing of the MERIT model than to misplacement of the stations. We find that 66.64 % and 79.57 % of the entire onshore database has a height error below 5 m and below 10 m compared to MERIT, respectively. The absolute values of the ENI database were referred to the old Potsdam gravity system and were transferred to the IGSN71 correcting the values for 14.00 mGal (Morelli, 1947; Wollard, 1979). In the areas with both ENI data and modern acquired data, the systematic shift was confirmed by direct comparison of the absolute gravity values.

The current published national gravimetric map of Italy and the adjacent seas was realized on a scale of 1 : 1 250 000 (APAT, 2005) using land and sea data extracted respectively from different databases as illustrated above in Sect. 1.2. All data were referred to the IGSN71 (International Gravity Standardization Net). This actual gravity map (Ferri et al., 2005) was compiled by the following parameters to be used for the land data: a constant nominal density of 2670 kg m$^{-3}$, the international formula 1980 (IAG 80) for normal gravity (Moritz, 1984), a second order Free Air reduction, a Bouguer correction calculating the effect of a spherical cap of surface radius 166.736 km, and a corresponding terrain correction extended to same distance using a digital elevation model.



### Slovenia

From the gravity map of the Geological Survey of Slovenia (Čibej, 1967) approximately 2150 gravity points were selected for the construction of the regional map at scale 1 : 100 000. Gauss-Krüger coordinate system was used and later transformed to WGS84. The average density of gravity points of this data set is 0.106 points per 1 km². The map was digitized and re-

interpolated between 1996 and 2000 by Stopar (2016). All gravity measurements were tied to the national gravity system which was linked to the Potsdam system. The average density of gravity points of this data set is 0.106 points per 1 km². In the original data set (Čibej, 1967) terrain corrections were computed up to the distance of 20 km. For the purpose of AAGRG compilation digital elevation models (DEM) for Slovenia in 12.5 m and 25 m grid sizes prepared from orthophoto surveys were used for terrain corrections. The general estimated accuracy of the model is 3.2 m, more specific: in flat areas 1.1 m, low

hills 2.3 m, medium hills 3.8 m and mountain areas 7.0 m (Surveying and mapping authority of Slovenia, 2019). Application of high resolution 1 m grid size DEM based on a recent LiDAR survey of the whole Slovenia was also considered.

In the frame of the W-E Europe Gravity Project leaded by Getech from Leeds a new dataset was prepared in 1990's which comprises 416 gravity points giving average density of 0.02 stations per 1 km² (Car et al., 1996). The Gauss-Krüger coordinate system was used and later transformed to MGI 1901 Bessel and WGS84. Datum and reference field was Potsdam 1967 in the

IGSN71 system with added atmosphere correction. Terrain corrections were computed up to the distance of 167.7 km using the density of 2670 kg m⁻³. The estimated accuracy of this data set is 0.05 mGal in flat areas and much lower in mountain areas.

### Switzerland

The Swiss Gravity Database GRAVI-CH was collected and maintained by the University of Lausanne (Olivier et al., 2010). It

consists of around 30 000 points with measurements from 1953 to 2000.

The data set used in this project is a subset of 7962 points from GRAVI-CH, limited to the area of Switzerland and Liechtenstein and reduced to a density of 1 point per 2 × 2 km point density extraction. Many of the Swiss gravity points have been measured on geodetic reference points. Their position accuracy is a few cm in the Swiss Projection System LV03. The positions of the other points have just been read from topographical maps 1:25 000. Their accuracy in position is in the order

of 10-20 m. All the data have been transformed to ETRS89 using the official method of the Federal Office of Topography. There is no further loss in positioning accuracy. The official height system of Switzerland LN02 uses just levelled heights without any gravity reduction. The height accuracy of the gravimetric points ranges from a few cm for triangulation or levelling benchmarks to 1-2 meters for points which were just taken from topographic maps. All these points were transformed to ellipsoidal heights in ETRS89 by using the official formulas of the Swiss Federal Office of Topography. A loss of accuracy in

the order of 10-20 cm is possible in rugged terrain. Most of the gravity points were originally observed in the old Potsdam gravity reference system but were transferred later into a modern system based on absolute gravity measurements.



In total, all these gravity data sets comprise 1 076 871 gravity stations. Figure 1 shows the spatial distribution of the original data sets country by country.


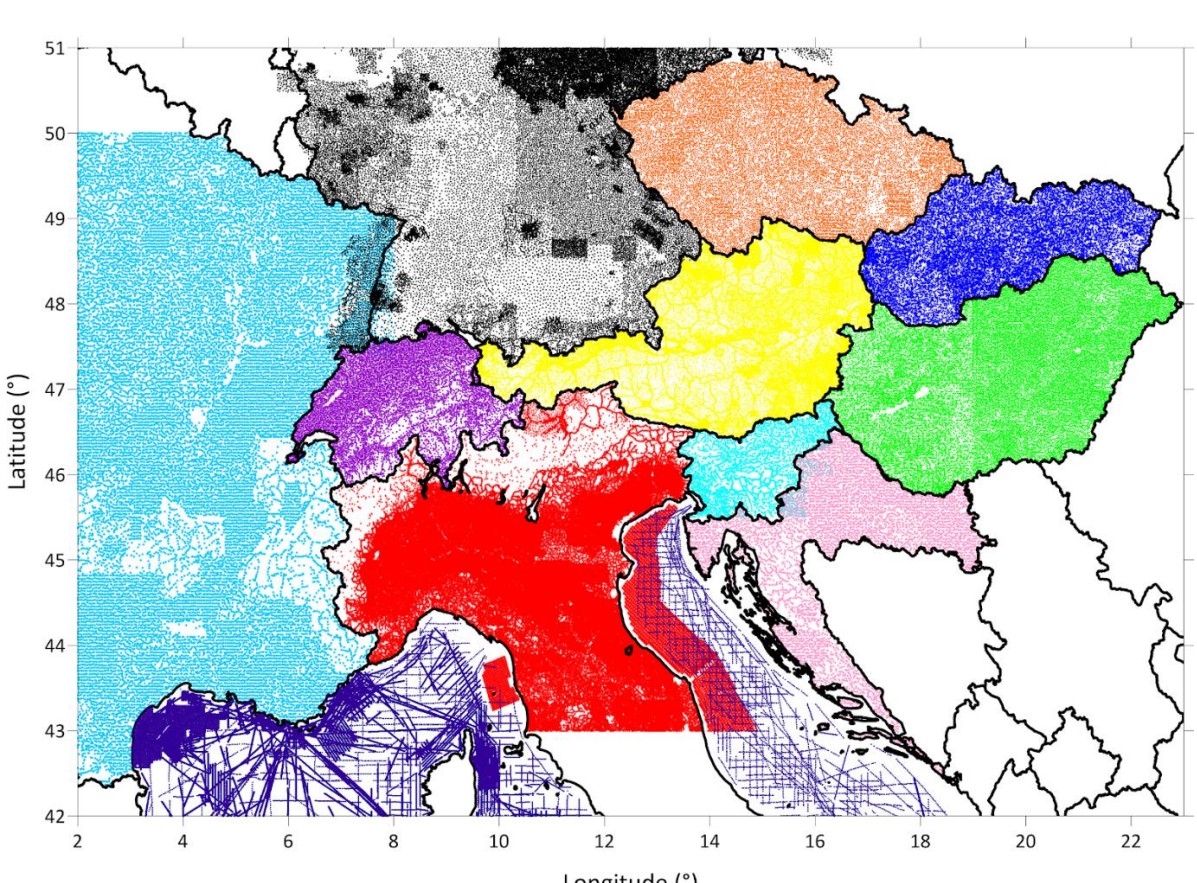

**Figure 1: The distribution of more than 1 million gravity stations in the area of investigation and compilation. Colours indicate the national databases used in the compilation. Starting on the left we designed: Light blue to France, black to Germany, orange to Czech Republic, dark blue to Slovak Republic, green to Hungary, pink to Croatia, red to Italy, dark blue to offshore areas, purple to Switzerland.**


## 2.1 Problems with positioning, heights, and gravity data

One of the key problems in the unification of gravimetric databases is the homogenization of position, height and gravimetric coordinate systems used in each database. Through its historical development, each country has used and sometimes still uses

local systems and their realisation (frame), which are often based on the established principles of reference systems using older ellipsoids or older geodetic reference networks and projections. These systems and their realisations thus contain several differences, which are responsible for large inhomogeneities, shifts, errors in position, height, and gravity. These errors are most evident in the mutual comparison of data from individual countries.



To avoid these problems in the position of gravimetric points, all position data were transformed from local systems to the
global system, i.e., the European Terrestrial Reference System 1989, which is accurate, homogeneous, and recommended for
all European countries (Altamimi, 2018; ETRS89, 2020). A similar situation is in the height systems where countries use
different types of physical heights, they are linked to different tide gauges and each country has a different practical
implementation of the relevant height system (EVRS, 2020). The solution is again transformation to a uniform platform in the
form of ellipsoidal heights in the ETRS89 system based on the ellipsoid GRS80 (Moritz, 2000). The situation is similar in
gravimetric reference systems, where especially the gravimetric databases that have been created for decades often use old
gravimetric systems connected to the Potsdam system. An important step was therefore to convert these data into gravimetric
systems, which are connected to absolute gravimetric points and measurements, such as IGSN71 (Morelli et al., 1972) or
modern national systems connected with the recent absolute measurements, which are verified by international comparisons
of absolute gravimeters (Francis et al., 2015).

For these transformations, national transformation services were used (operated by national mapping services e.g., SAPOS,
SKPOS) or transformations implemented into standard GIS tools or our own software implementations based on national
standards, information, and experience of individual responsible institutions. The transformation from physical heights in
national vertical systems to ellipsoidal heights in the ETRS89 system, ellipsoid GRS80 was realized using available local
geoid/quasigeoid models available through transformation services or implemented in current geodetic processing programs
(e.g., Trimble Business Center, Leica Infinity). If a local geoid/quasigeoid model was not available for some areas, then the
global geopotential model EIGEN 6C4 (Förste et al., 2014) was used for transformation. This model was also used for marine
data, where the height of points was not given or had zero value.

Provided data include a local identifier, horizontal coordinates in the local coordinate systems (except France and Croatia),
physical height, ellipsoidal coordinates in the ETRS89 system, ellipsoidal height above the GRS80 ellipsoid (except France,
the Czech Republic and Slovenia) and the gravity value. For each parameter available metadata describing e.g., coordinate
system (ellipsoid, EPSG code), used transformation method or transformation service, local geoid/quasigeoid were also
collected.

For datasets where all information was available, an independent transformation control check was performed between the
local and global coordinate system, respectively between physical and ellipsoidal heights using available geodetic
geoid/quasigeoid models. Differences in position were in the majority of cases less than 1 m. All larger differences were
individually investigated. A similar situation was for the heights, where differences were generally less than 50 cm. These
differences were mostly caused by different transformations, its practical software realization, or local specifics of the dataset.
Figure 2 shows the transformation scheme. Data statistics and an overview of selected metadata are given in Table 1.





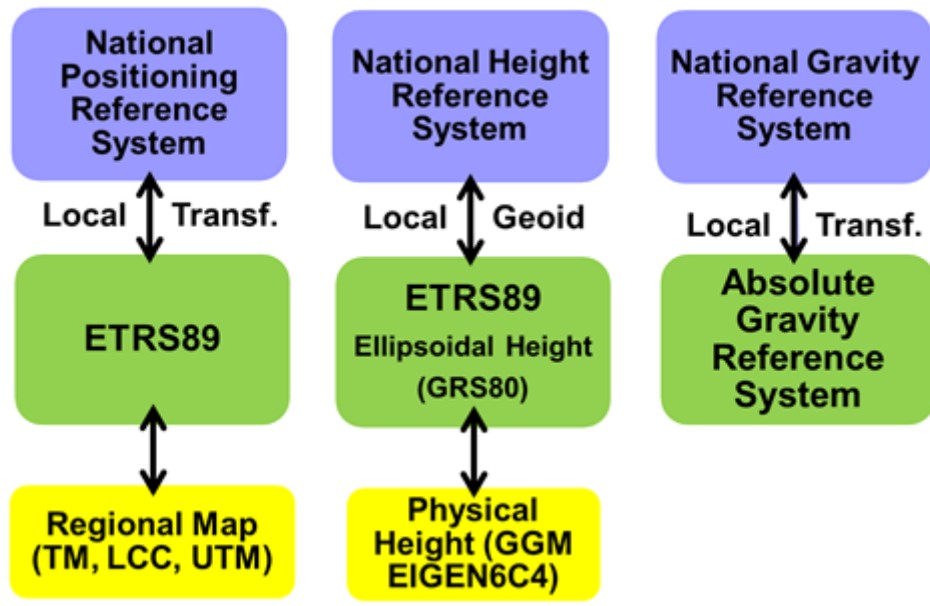


**Figure 2: Transformation scheme for unification of the national positioning, height, and gravity reference systems.**

| | Quantity of points | Position | | Height | | | Gravity | Notes and references |
|---|---|---|---|---|---|---|---|---|
| | all / used | National | ETRS89 | Physical | Geoid/ Quasi-geoid | Ellip-soid | g value | |
| Austria | 54 251/ 51 811 | MGI | x | Trieste | BEV GV 2008 | x | x | Pail et al., 2008 |
| Croatia | 4939/ 4565 | - | x | Trieste HVRS1971 | HRG2009 | x | IGSN71 | Basic and Bjelotomi, 2014 |
| Czech Republic | 13 955/ 13 831 | Krovak S-JTSK | x | Kronstadt Baltic height system | CR-2005 | - | ABS Sgr95 | Kostelecky et al., 2004 |
| France | 58 750/ 57 889 | - | x | Marseille | RAF09 | - | x | IGN, 2010 |
| Germany | 36 442/ 36 440 | UTM32 | x | Amsterdam DHHN | GCG2016 | x | IGSN71 | Schwabe et al.,2016 |



| Hungary | 25 434/ 25 147 | EOV | - | Kronstadt Baltic height system | VITEL2014 | x | ABS MGH-2000 | VITEL, 2020 |
|---|---|---|---|---|---|---|---|---|
| Italy | 132 074 30 821 | - | x | Genoa | ITALGEO05 | x | Potsdam/I GSN71 | Barzaghi et al., 2007 |
| Slovakia | 21 108/ 21 108 | Gauss-Krüger S-42 | x | Kronstadt Baltic height system | DVRM05 | x | ABS Sgr95 | Zahorec et al., 2017 ZBGIS, 2020 |
| Slovenia | 3066/ 364 | Gauss-Krüger D-48 | x | Trieste | SLOAMG2000 | - | IGSN71 | Kuhar et al., 2011 |
| Switzer-land | 7962/ 7962 | Oblique Mercator LV03 | x | Marseille LN02 | CHGEO04A | x | ABS LSN2004 | Olivier et al., 2010; Marti, 2007 |
| Former Yugoslavia | | Gauss-Krüger S-42 | - | | EIGEN6C4 | | CBA | Bilibajkic et al., 1979 |
| Marine | 718 890/ 716 402 | | x | zero | EIGEN6C4 | | FA | |


**Table 1: Data statistics and an overview of selected metadata. From the total of originally 1 076 871 gravity stations 1 066 340 data were used for the compilation of the gravity maps.**

### 2.2 Digital elevation models

One of the important elements in the CBA calculation process is the determination of mass correction (MC). The key element for quality and reliable determination of MC is the use of reliable and accurate digital terrain models without canopy and buildings. Since our approaches to MC are based on calculations in different zones (Sect. 3), it is very important to provide models with the appropriate resolution and quality. The nearest zone T1, up to 250 m, is the most critical from the MC point of view. Hence, for this zone, it is best to use the highest quality models based on LiDAR technology, respectively Digital

photogrammetry with 1-10 m resolution. Each country, depending on availability, provided a model suitable for calculating the TOPOSK software "inner zone T1" (Sect. 4.1). Basic metadata summary is in Table 2. Acquired models differ in the raw data collection methods, resolution, time of creation, position and height coordinate system, accuracy. Due to the problem of coordinate systems unification (especially height system) and general approach to MC calculation, the heights in all models were transformed to ellipsoidal heights in the ETRS89 system, ellipsoid GRS80 using the appropriate local geoids/quasigeoids

of the individual countries.





| | Source | Grid step (m) | Reference |
|---|---|---|---|
| Austria | L DGM10 Österreich Geoland | 10 | http://www.geoland.at |
| Croatia | MERIT | 25 | http://hydro.iis.u-tokyo.ac.jp/~yamadai/MERIT_DEM/ |
| Czech Republic | L DMR5G-V CUZK | 5 | https://geoportal.cuzk.cz/ |
| France | L/SRTM DTM France Sonny | 20 | http://data.opendataportal.at/dataset/dtm-france |
| Germany | L DGM10 BKG | 10 | http://gdz.bkg.bund.de/ |
| Hungary | TM DDM BFKH | 30 | http://www.ftf.bfkh.gov.hu/ |
| Italy | MERIT | 25 | http://hydro.iis.u-tokyo.ac.jp/~yamadai/MERIT_DEM/ |
| Slovak Republic | TM DMR3.5G GKU | 10 | https://www.geoportal.sk/en/ |
| Slovenia | P/L LIDAR ARSO | 12.5 | http://www.geoportal.gov.si/eng/ https://gis.arso.gov.si/ |
| Switzerland | L swissALTI3D SwissTopo | 5 | https://www.swisstopo.admin.ch/ |

**Table 2: List of DEMs used for test and mass correction calculations in the "most inner zone" of the TOPOSK program (Sect. 4.1) of the individual countries; the grid spacing, sources and internet references are given. The letters stand for the techniques used in** 545 **the DEM compilation: "L" for LIDAR, "P" for Photogrammetry, "TM" for heights from digitized topographic maps, and "MERIT" and "SRTM" for the radar data.**

Each of these models was tested on a set of gravimetric points located at least 500 m from the border of each country. This test served both to detect possible artefacts in the DEMs (especially in high mountain areas) and also as a primary filter of the 550 quality of the position of gravimetric points. These differences are illustrated in Fig. 3 and statistical findings in Table 3. Several points exceeding the threshold of ±50 m of difference between the measured and interpolated height were separately assessed and subsequently excluded from the database. The biggest differences are in Slovenia and the mountainous parts of France, most likely due to the poor quality of station positions of old gravity data. Fig. 4 presents the frequency distribution of the height residuals for the data sets of all contributing countries.


| | Austria | Croatia | Czech-Repub. | France | Germany | Hungary | Italy | Slovak-Repub. | Slovenia | Switzer-land |
|---|---|---|---|---|---|---|---|---|---|---|
| Nr. points (m) | 51 381 | 4565 | 13 626 | 57 248 | 34 702 | 24 894 | 110 664 | 21 108 | 326 | 7628 |
| Minimum (m) | -32.12 | -49.98 | -49.42 | -49.91 | -19.61 | -30.05 | -49.97 | -45.46 | -45.83 | -44.65 |
| Maximum (m) | 72.40 | 49.56 | 49.85 | 49.66 | 10.09 | 33.92 | 49.98 | 39.01 | 47.85 | 33.38 |
| Mean (m) | 0.14 | -0.56 | 0.39 | -1.09 | -0.04 | 0.75 | 0.29 | 0.28 | -0.57 | 0.25 |
| Standard deviation (m) | 2.06 | 13.85 | 8.06 | 8.58 | 1.48 | 3.16 | 10.34 | 5.22 | 17.28 | 2.58 |

**Table 3: Statistical results of test calculations of consistency of surface station heights and used DEMs of the individual countries in the "most inner zone".**






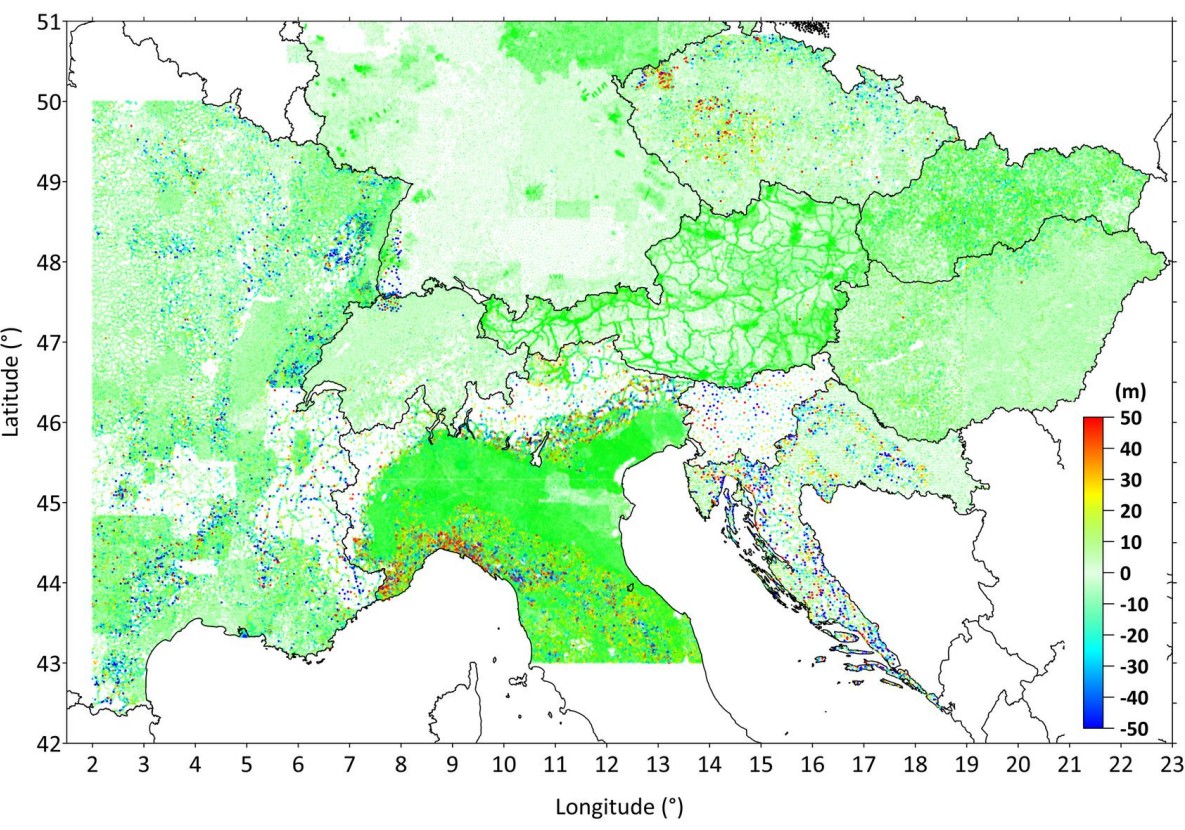

**Figure 3: Height differences (in meters) for gravity stations in the "inner zone" of the TOPOSK software (refer to Sect. 4.1) between the used DEMs and the heights of these stations.**

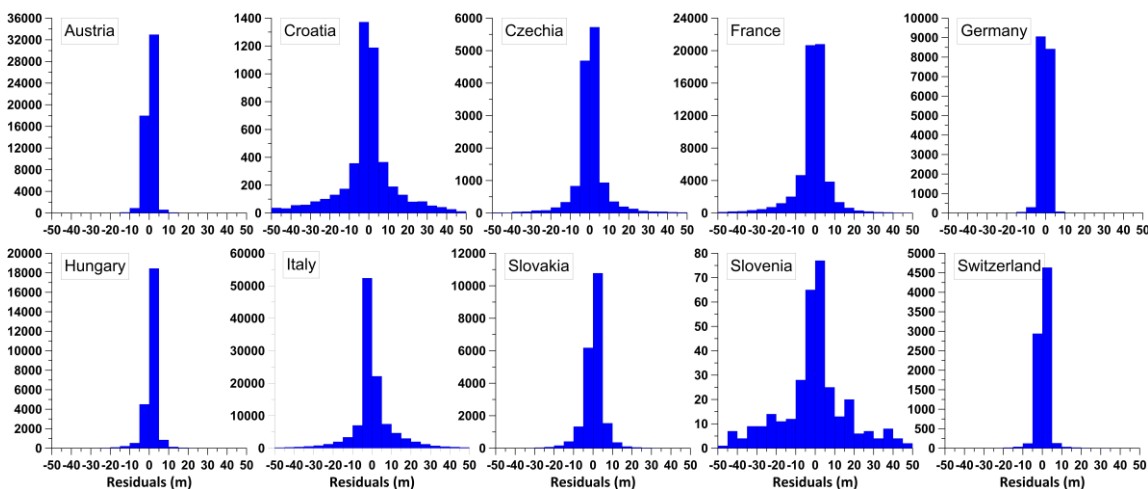

**Figure 4: Histograms of height difference residuals of participating countries. The values in the different classes are given in meters.**



For the calculation of MC within the middle zone (250 m - 5240 m) it is very suitable to use DEMs with medium resolution (1 - 3 sec), which uniformly cover the whole territory, have the same shape representation, accuracy and can be converted with local geoid/quasigeoid models to ellipsoidal heights. Thanks to remote sensing satellite techniques, several commercial or freely available digital elevation models are currently available (https://insitu.copernicus.eu/library/reports/OverviewofGlobalDEM_i0r7.pdf). We analyzed the mostly used and freely available models: Advanced Land Observing Satellite World 3D 30 m version 2.1 (AW3D30, Tadono et al., 2014; Takaku et al., 2018), Advanced Spaceborne Thermal Emission and Reflection Radiometer (ASTER) Global Digital Elevation Model version 3 (ASTER GDEM, ASTER, 2020), NASA Shuttle Radar Topography Mission Global 1 arc second (SRTMGL1, NASA JPL 2013), Multi-Error-Removed Improved-Terrain DEM (MERIT DEM, Yamazaki et al., 2017) and Digital Elevation Model over Europe version 1.1 (EU-DEM, EU-DEM, 2017). All models (Table 4) represent a digital surface model (with urban and canopy artefacts), only the MERIT model has partially removed vegetation and represents a mix of a digital surface and terrain model.

| Model | Horizontal resolution (m) | Vertical accuracy (m) | Reference |
|---|---|---|---|
| ALOS AW3D30 | 30 | 7 | Tadono et al., 2014; Takaku et al., 2018 |
| ASTER GDEM | 30 | 15-20 | ASTER, 2020 |
| EU-DEM | 25 | 5-7 | EU-DEM, 2017 |
| MERIT | 90 | 5-12 | Yamazaki et al., 2017 |
| SRTMGL1 | 30 | 6-9 | NASA JPL, 2013 |

**Table 4: Basic characteristics of the tested global DEMs.**

From these models the best one is MERIT due to the removal of major error components from the satellites DEMs like absolute biases, stripe, speckle noise and canopy height biases (Yamazaki et al., 2017; Hirt 2018). This was confirmed also by an independent comparison at selected gravimetric points with new exactly measured position with GNSS in Switzerland, Slovenia, and Slovakia (refer to Table 5 and Fig. 5), where large errors in the mountainous parts were due to canopy. MERIT DEM was used in the original 3 arcsec resolution and for T2 zone calculation it was resampled to the 25 m resolution.

The overall quality of the MERIT model has been tested at most gravity station heights. The differences can be seen in Fig. 6 and basic statistical data in Table 6.





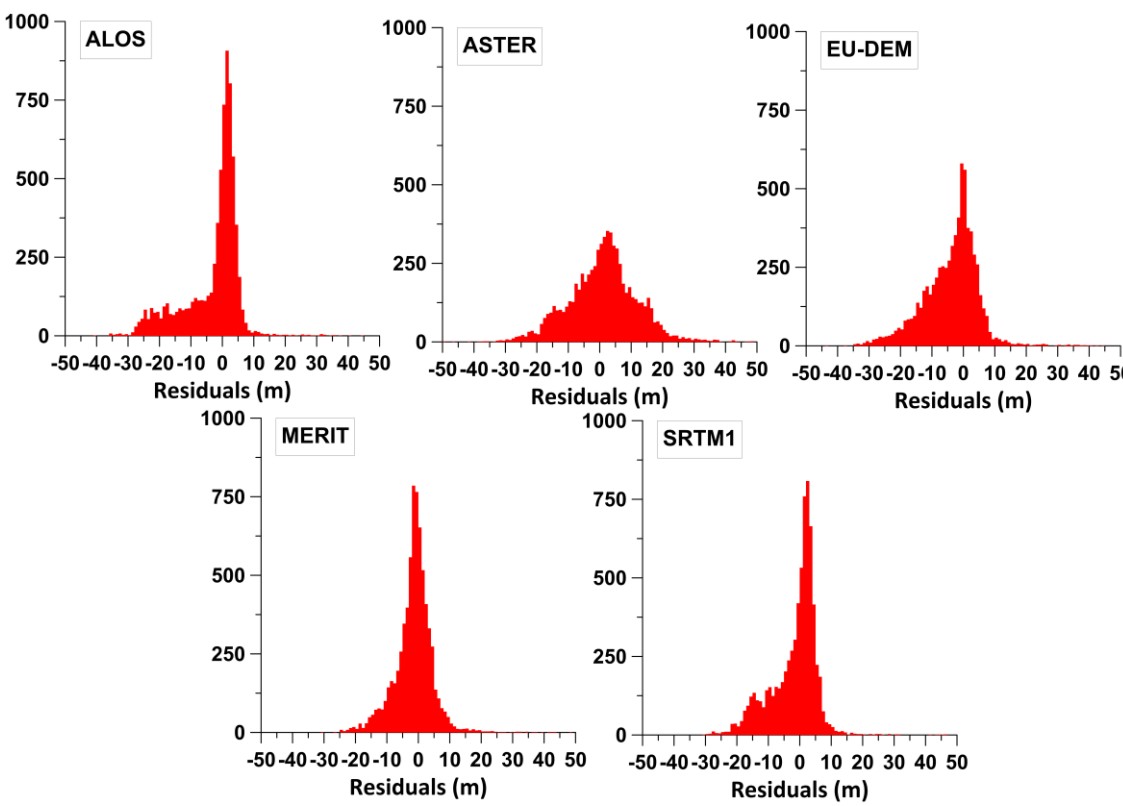


**Figure 5: Histograms of height residuals between global DEMs and 7 097 selected gravity stations on the territory of Slovakia. The values in the different classes are given in meters.**


|  | ALOS | ASTER | EU-DEM | MERIT | SRTM1 |
|---|---|---|---|---|---|
| Minimum (m) | -40.35 | -49.09 | -43.60 | -30.88 | -30.08 |
| Maximum (m) | 181.45 | 186.17 | 117.17 | 75.53 | 183.06 |
| Mean (m) | -2.83 | 1.07 | -3.83 | -1.43 | -1.63 |
| Standard deviation (m) | 9.28 | 11.30 | 9.25 | 6.23 | 7.74 |

**Table 5: Statistical results of test calculations of consistency of station heights on the territory of Slovakia (7097 points) and tested global DEMs.**




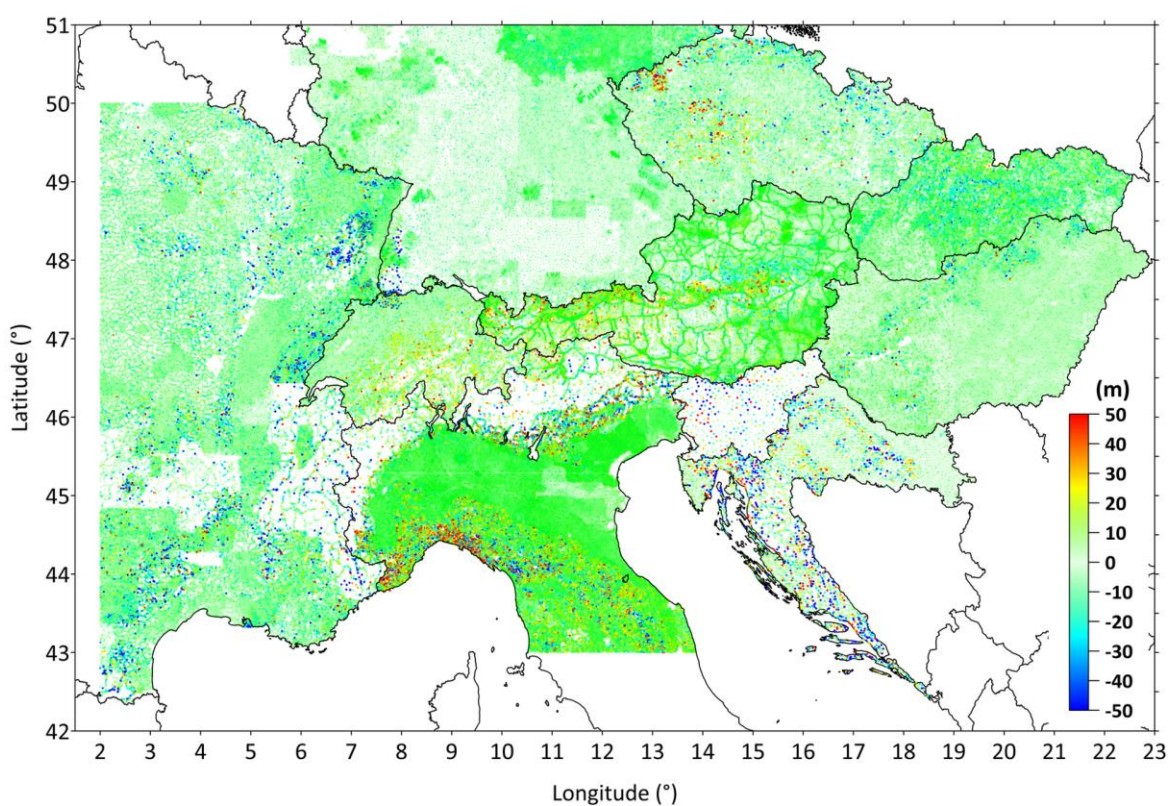

**Figure 6: Height differences (in meters) between MERIT DEM heights and heights of original surface gravity stations; MERIT DEM heights were considered for the "middle zone" of the mass calculation software TOPOSK (refer also to Sect. 4.1).**


|  | Austria | Croatia | Czech Republic | France | Germany | Hungary | Italy | Slovak Republic | Slovenia | Switzer-land |
|---|---|---|---|---|---|---|---|---|---|---|
| Points | 51 678 | 4939 | 13 955 | 58 750 | 36 442 | 25 434 | 110 664 | 21 108 | 416 | 7962 |
| Minimum (m) | -87.77 | -944.20 | -172.48 | -250.78 | -38.52 | -260.18 | -49.97 | -60.91 | -179.48 | -70.31 |
| Maximum (m) | 126.33 | 253.37 | 305.81 | 243.12 | 28.45 | 112.85 | 49.98 | 44.11 | 103.16 | 96.70 |
| Mean (m) | 0.00 | -4.96 | -1.46 | -3.67 | -2.55 | -0.74 | 0.29 | -2.79 | -6.05 | 2.04 |



| Standard deviation (m) | 6.87 | 39.56 | 11.41 | 13.15 | 3.83 | 5.22 | 10.34 | 7.64 | 32.27 | 8.85 |
|---|---|---|---|---|---|---|---|---|---|---|

**Table 6: Statistical results of test calculations of consistency of station heights and used MERIT DEM.**

Largest differences were observed in Croatia, Czech Republic, France, and Hungary most likely due to the low quality of the position of gravity stations.

**2.3 Bathymetry and lake depths data**

When calculating bathymetric corrections (BC), the gravity effect is calculated due to the difference in density between the water masses of the offshore areas and those of the land masses. In contrast to the MC, we calculate BC with physical heights. The reason is explained in Fig. 7. Water masses above the ellipsoid level are thus taken into account with their real density of 620 1030 kg m$^{-3}$. We used a detailed bathymetric model EMODnet (2018) with the resolution of 3.5 sec. A harmonised DEM has been generated for European offshore regions from selected bathymetric survey data sets, composite DTMs, Satellite Derived Bathymetry (SDB) data products, while gaps with no data coverage were completed by integrating the GEBCO Digital Bathymetry.

**3 Numerical background for complete Bouguer anomaly (CBA) calculation**

The main output of the recalculation and homogenisation of the gravity data sets is the computation of complete Bouguer anomaly values (CBA). Basic version of the CBA from the studied region was calculated for ellipsoidal heights of calculation points with geographical coordinates ($\lambda$, $\varphi$), with mass corrections (terrain effects of masses) $\delta g_M$ extending to the standard distance of 166.7 km, with bathymetric corrections $\delta g_B$ and simplified atmospheric corrections $\delta g_A$. In contrast to the conventional processing of Bouguer gravity, where the gravity effects of the spherical Bouguer plate and topography are 630 calculated, we have performed a mass correction for all masses between the ellipsoidal reference surface and the physical surface. Mass corrections were calculated by means of the Toposk software (refer to Sect. 4), using ellipsoidal heights $h_E$ of the calculation points and ellipsoidal digital elevation models (using in majority local geoids for the transformation). A control on selected groups of points was performed with the TriTop software (Holzrichter et al., 2019, details in Sect. 4). Also, the Free-Air corrections were calculated for ellipsoidal heights. On the other hand – the bathymetric and simplified atmospheric 635 corrections were calculated for physical heights $H$ of the calculation points (explanation in Sect. 4 and Fig. 7). Bathymetric corrections were also calculated by means of the Toposk software, but in a slightly adjusted mode (see below).





**Basic formula for the CBA calculation was adopted from Meurers et al. (2001):**

$$BA(\lambda, \varphi, h_E) = g(\lambda, \varphi, h_E) - \gamma(\varphi, h_E) - \delta g_M(\lambda, \varphi, h_E) + \delta g_B(\lambda, \varphi, H) + \delta g_A(\lambda, \varphi, H) \tag{1}$$

$$\gamma(\varphi, h_E) = \gamma_0(\varphi) + \frac{\partial \gamma}{\partial h_E}\Big|_0 h_E + \frac{1}{2}\frac{\partial^2 \gamma}{\partial h_E^2}\Big|_0 h_E^2 \tag{2}$$

where $\gamma_0(\varphi)$ results from the well-known Somigliana formula for the normal gravity acceleration of a rotational ellipsoid at its surface (Somigliana, 1929; Heiskanen and Moritz, 1967):

$$\gamma_0(\varphi) = \frac{a\,\gamma_E \cos^2\varphi + c\,\gamma_P \sin^2\varphi}{\sqrt{a^2 \cos^2\varphi + c^2 \sin^2\varphi}} \tag{3}$$

and higher vertical derivatives of $\gamma(\varphi, h_E)$ are given by:

$$\frac{\partial \gamma}{\partial h_E}\Big|_0 = -\frac{2\gamma_0}{a}\left(1 + f - 2f\sin^2\varphi + \frac{3}{2}f^2 - 2f^2\sin^2\varphi + \frac{1}{2}f^2\sin^4\varphi\right) - 2\omega^2 \tag{4}$$

$$\frac{\partial^2 \gamma}{\partial h_E^2}\Big|_0 = \frac{6\gamma_0}{a^2\left(1 - f\sin^2\varphi\right)^2} \tag{5}$$

All constants in Eq. (3) to Eq. (5) were taken from the Geodetic Reference System 1980 (GRS80), e.g., in Moritz (1984):

    $\gamma_E$ = 9.780 326 771 5 m s$^{-2}$, normal gravity acceleration at equator,
    $\gamma_P$ = 9.832 186 368 5 m s$^{-2}$, normal gravity acceleration at pole,
    $a$ = 6 378 137 m, semi-major axis of the normal ellipsoid,
$c$ = 6 356 752.314 1 m, semi-minor axis of the normal ellipsoid,
    $f$ = 0.003 352 810 681 18, geometrical flattening,
    $\omega$ = 7.292 115×10$^{-5}$ rad s$^{-1}$, angular velocity of the Earth's rotation.

Simplified atmospheric corrections $\delta g_A$ (Wenzel, 1985) were calculated by means of the approximation:

$$\delta g_A(\lambda, \varphi, H) = 0.874 - 9.9\times10^{-5}H + 3.56\times10^{-9}H^2 \quad (\delta g_A \text{ in mGal, } H \text{ in meter}) \tag{6}$$

**4 Reprocessing of original data and corrections**

One of the main problems in the homogenisation of data and recompilation of gravity fields was the use of different procedures for the calculation of mass correction (MC) and bathymetry correction (BC) by the national operators/authorities. This meant that a complete recalculation had to be carried out for the new compilation based on the available point data. We will come back to this in detail in Sect. 4.2 and Sect. 4.3. Another special feature was the calculation of the gravimetric effects of the

Alpine lakes on the basis of bathymetric data of the region (Sect. 4.4).





An important first step before starting the recompilation was to test and select the available software to calculate the mass corrections. We compared two custom software packages developed by team members - Toposk and TriTop (Sect. 4.1). A special approach to reprocessing required the use of a digitized old CBA map from the former SFR Yugoslavia (Bilibajkič et al., 1979) (Sect. 4.5).

Further improvements of the new CBA map are the refined calculations of an atmospheric correction and the future containment of distant terrain/bathymetry effects (Sect. 4.6).

From the methodological viewpoint, the use of ellipsoidal heights for CBA calculation is innovative. Considering the participating countries, so far this concept has only been used in Austria (Meurers and Ruess, 2009). It ensures that Bouguer anomalies, which then, in the sense of physical geodesy, actually are gravity disturbances corrected for terrain mass effects,

are not disturbed by the geophysical indirect effect (GIE, e.g. Li and Götze, 2001; Hackney and Featherstone, 2003) contrary to Bouguer anomalies relying on physical heights If the normal field in Eq. (1) is defined at the height above the surface ellipsoid, it is necessary to define the effects of terrain/bathymetry masses above the ellipsoid (not above the geoid). Therefore, the concept requires the use of ellipsoidal heights of the observation points and at the same time it is necessary to transform the topography/bathymetry grids from physical to ellipsoidal heights. In the AlpArray area the situation is more or less simple,

the ellipsoid is below the geoid throughout the region (approx. 30 to 55 m). This greatly simplifies the calculation. In the case of continental areas, we get a slightly thicker layer of topography, whose effect is calculated in the same way as in the case of physical heights (with the density of 2670 kg m$^{-3}$). In the case of marine areas, the situation is somewhat more complicated as partly the ocean masses are above the ellipsoid level. If we want to take these into account with their real density (1030 kg m$^{-3}$), it is necessary to separate their effect from terrain masses. Numerically, this can be done by taking these water masses into

account first as topographic masses (i.e., with a density of 2670 kg m$^{-3}$) and then also as part of the bathymetric correction (i.e., with a density of -1640 kg m$^{-3}$), however, now counted as in the classical concept of physical heights (Fig. 7.). As a result, we assign a density $\rho$ of 1030 kg m$^{-3}$ to these water masses ($\rho$ = 2670 kg m$^{-3}$ − 1640 kg m$^{-3}$).

In connection with the above calculation methods, one note is appropriate. The difference between the two versions (physical vs. ellipsoidal heights) of the CBA defines GIE, which has a normal gravity component (defined by the Free-Air gradient) and

a component defined by the gravitational attraction of the masses between the geoid and the ellipsoid. In our case, this second component is equal to the total gravitational effect of these masses with a density of 2670 kg m$^{-3}$ (no difference in density at sea and on land). This is in apparent contradiction to published papers which state that the GIE should be calculated with different densities for land and sea (offshore with a density of 1030 kg m$^{-3}$). This apparent discrepancy is due to different approaches to bathymetric correction. The approach of Chapman and Bodine (1979) is based on Free-Air anomalies which do

not include bathymetric corrections, unlike our CBA. The GIE is thus easier to define in our case (for a constant density of 2670 kg m$^{-3}$ in the whole considered space between the geoid and the ellipsoid), thanks to the consideration of the rock-water density contrast in this space as part of the bathymetric correction.





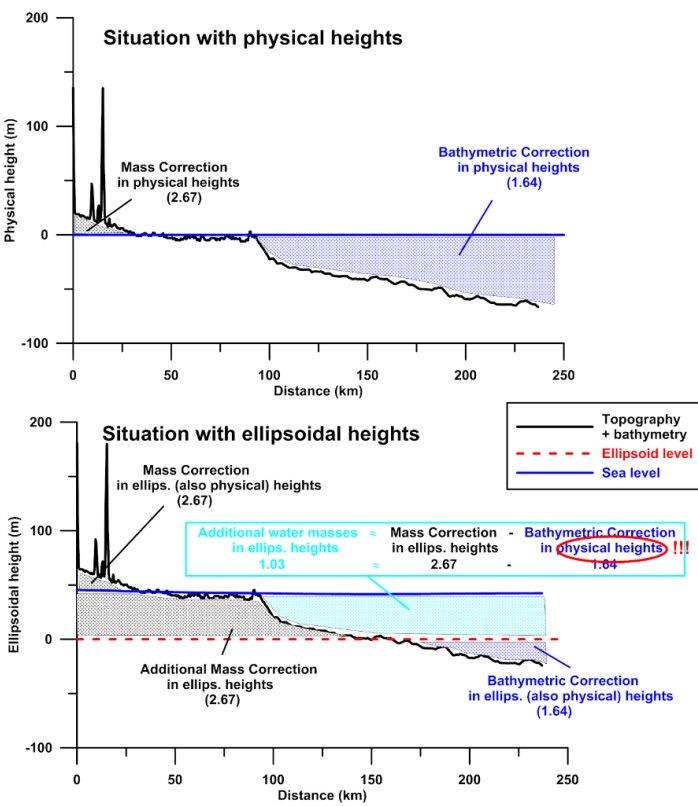

**Figure 7: Schematic comparison of ellipsoidal vs. physical concept of CBA. Note that the effect of additional water masses is calculated in a two-step process.**

## 4.1 The software test for calculations of mass correction

The Toposk software (Zahorec et al., 2017) is designed for the calculation of the gravitational effect of the near terrain masses

for both "near terrain effect" (NTE) and "mass correction" (MC), i.e., the total masses between the topography and the zero

level - geoid or ellipsoid (we point out the difference from the terrain correction (TC), which represents only masses exceeding

the classical "Bouguer shell"). The program is suitable for highly accurate calculations in rugged terrain using high-resolution

DTMs. Different DTMs, with increasing resolution towards the calculation point, are used within particular zones. By default

the program uses the following zoning:

705          T1: inner zone (0 - 250 m from the calculation point),

          T2: intermediate zone (250 - 5240 m) and

          outer zones: T31 (5.24 - 28.8 km) and T32 (28.8 - 166.7 km).

The standard outer limit of 166 730 m (equivalent to the spherical distance of 1°29′58″) represents the outer limit of the zone

$O_2$ of the Hayford-Bowie system. Different analytic formulas are used within particular zones. 3D polyhedral bodies are used





within the inner zone. The planar approach is applied within the inner and intermediate zones, leading to a small negligible error with maximum of a few tens of μGal for a density 2670 kg m$^{-3}$ (Zahorec et al., 2017). The outer zones are treated by a spherical approach. By default, for the inner zone, the height used for the calculation of the correction at the position of gravity station is interpolated from the DTM in order to reduce errors resulting from the height mismatch between point and DTM.

The TriTop software (Holzrichter et al., 2019) is an adaptive algorithm for MC based on a triangulated polyhedral

representation of the topography. The runtime of the algorithm is improved by an automatic resampling of topography. The topography is resampled in a quadtree structure. High resolution of the topography is only considered if it has a significant influence on the gravitational effect at the station and not only by the distance to the station. Therefore, there are no default zone radii definitions, but the resolution depends only on the gravitational effect and differs for each station. In comparison to Toposk, Tritop does not consider a high resolution zone (T1, see above) and does not interpolate topography in this zone in

dependence to station height. The DTM heights are not modified.

The programs were compared to each other on different sets of points from Slovakia and Austria. Mainly the second comparison was important, because of the typical Alpine terrain character of the majority of the territory in Austria. The obtained results by the Toposk and TriTop software were compared with previously computed mass corrections (NTE) from the Austrian gravity database. This comparison was realized on a set of 28 420 points with the ellipsoidal heights ranging from

158.35 m to 2898.78 m. The character of the differences between mass corrections from the Austrian gravity database and NTE calculations by means of programs Toposk and TriTop is visible from histograms in Fig. 8. Finally, the Toposk software was selected for recalculation of MC effects due to better statistical parameters (median and standard deviations) and the absence of outliers in the calculations. The differences in MC of both algorithms are observed in areas where stations are located close to steep slopes in topography. The differences of the results in Austria are caused by the main difference of both

algorithms, and in particular the handling of the inner zone T1. TriTop does not change or interpolate the topography around the station. This might lead to larger correction values in areas of highly rugged terrain due to steep slopes close to the station or even in cases in which the station height is slightly below the DTM. The comparison shows that in the area of highly rugged terrain the inner zone just around a station should be handled separately from the rest. Therefore, we decided to perform mass corrections by the Toposk software.






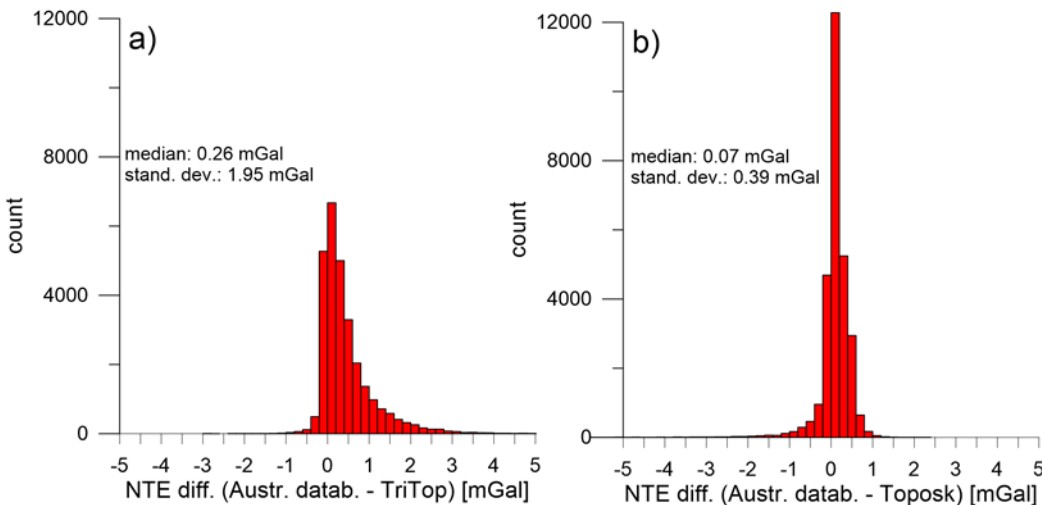

**Figure 8: Comparison of the differences between original mass corrections from the Austrian gravity database and NTE calculations**
**by means of programs (a) TriTop and (b) Toposk.**

## 4.2 Mass correction

For most countries, we used the available local detailed DEMs (refer to Sect. 2.2) with the resolution of 10-20 m (derived

mainly from LiDAR data) for calculation in the innermost Toposk zone (T1). For all other zones we chose the best available

global DEMs. We got good results with SRTM models for outer zones. For the intermediate zone T2, we decided to use the

MERIT model based on our tests (Sect. 2.2). MERIT was also used for the inner zone if local models were unavailable. This

model (resampled to a 1 sec resolution) showed better height accuracy compared to other global models (based on the height

residues at the points of the databases tested) and consequently minor differences in MC compared to local models (Fig. 9).

The mentioned height residues of individual points of the databases in relation to local (or MERIT) models, were subsequently

used as a control criterion. In particular, we consider points with height residues greater than ±50 m to be untrustworthy and

they were excluded from the CBA compilation process. The following graphs and maps are compiled without these excluded

points.



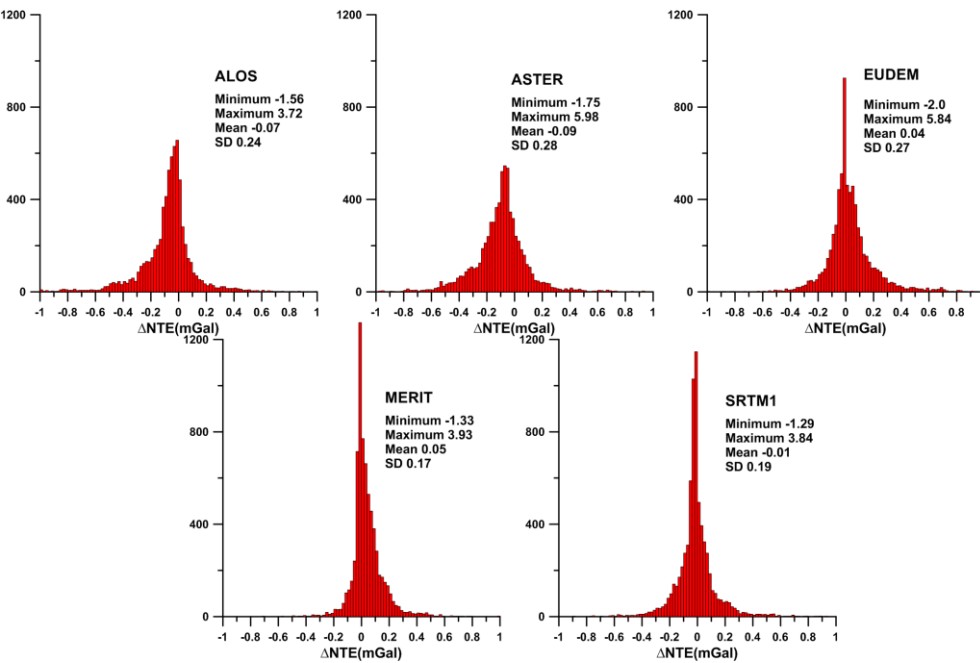

**Figure 9: Near terrain effect (or mass correction, density 2670 kg m⁻³) differences calculated using various global models compared to the local Slovak terrain model DMR-3. The test was made on approx. 8000 points covering the whole territory of Slovakia.**

Fig. 10 shows the MC values at all collected points. They reach values up to 375 mGal, while the ellipsoidal height of the points is from about 35 to 3938 m. The height dependence of the calculated MC is displayed in the lower right corner of the

figure. The difference between the calculated MC and the gravitational effect of the truncated spherical layer (to the same distance) defines classic terrain corrections. They reach values of almost 100 mGal.

There are options to verify calculated MC values and estimate their error. For some databases, we had the original MC or TC values, which allows us to compare and control different approaches. Fig. 11 shows graphs and statistical comparisons for some countries. The maximum differences are at the level of several mGal, the RMS error in most cases is below 1 mGal.

Note that the graphs do not show excluded points (above ±50 m height criterion), where significant differences in MC may be obtained. Another possibility to estimate the accuracy of the calculated MC is to compare the MC from the inner zone (where we can expect the most significant errors) for local DEMs and MERIT models. Fig. 12 shows a map of these differences. The maximum differences are locally at the level of a few mGal and are mainly bound to mountain areas.



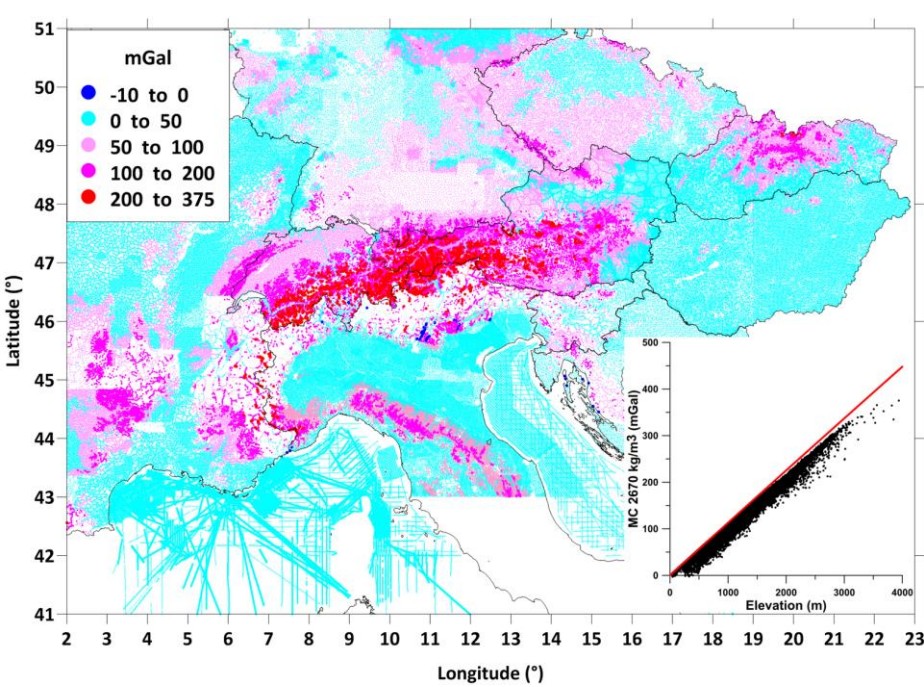


**Figure 10: Map of mass correction (up to the distance of 166 730 m, density 2670 kg m⁻³). Note the negative values of several mGal for a few points (dark blue points), which are mainly in deep valleys and near the coast. The graph in the bottom right corner shows the height dependence of the calculated MC. The red line represents the gravitational effect of the truncated spherical layer (up to the distance of 166.7 km, density 2670 kg m⁻³) for comparison.**


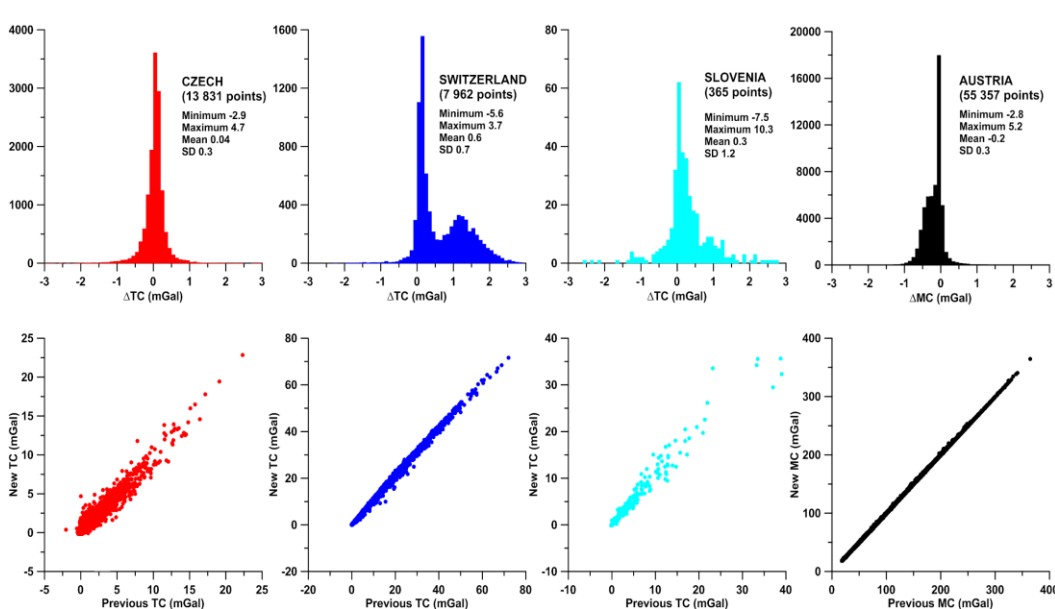

**Figure 11: Comparison of original mass correction (or terrain corrections) values and values calculated using local DEMs. Note: There are different scales for each graph.**

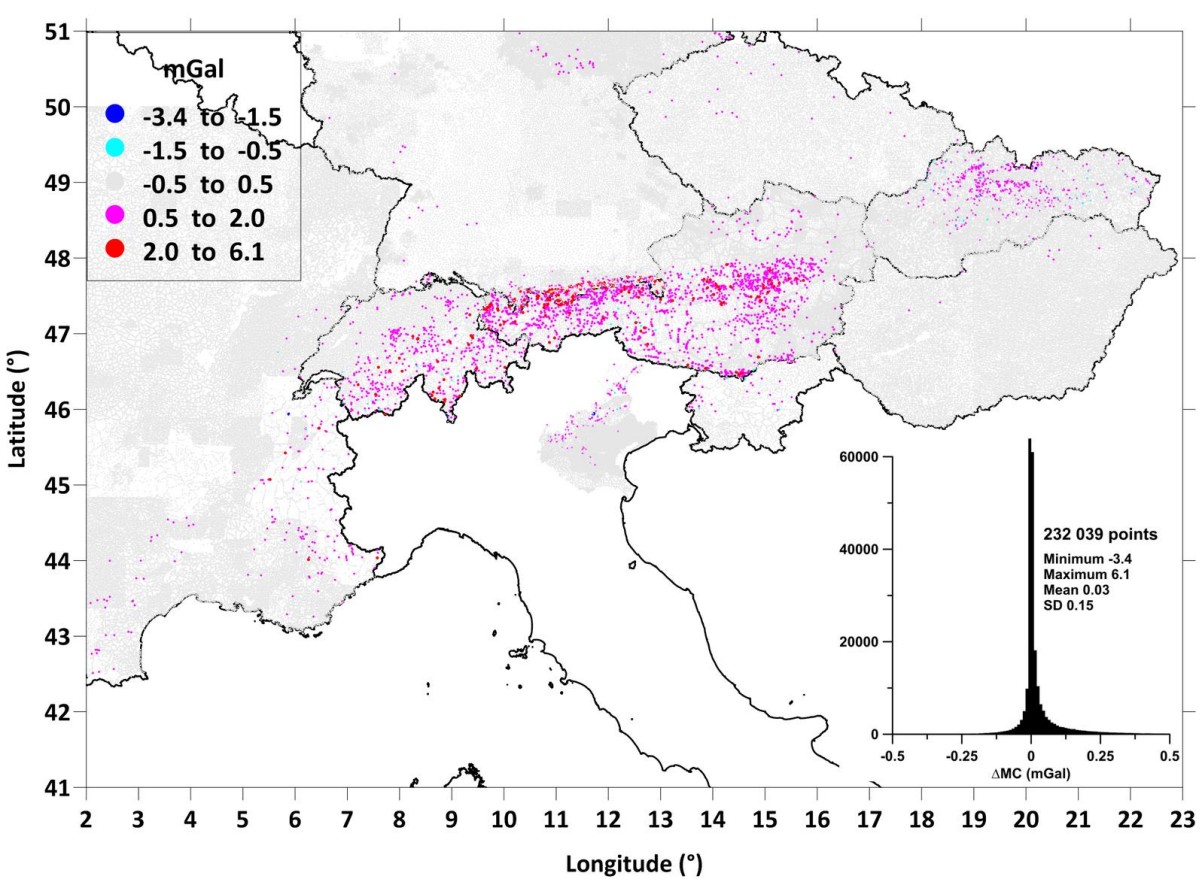

**Figure 12: Differences in mass correction values (correction density 2670 kg m⁻³) calculated by local DEMs which are derived mainly from LiDAR data and the MERIT model. For Italy, the part of the territory is displayed where for test reason a local high-resolution DEM was used.**

## 4.3. Bathymetric correction

Bathymetric corrections reach significant values for offshore and near coastal points and amount to more than 200 mGal (Fig. 13). The comparison with the frequently used planar approximation is in the upper right corner of the figure. Unlike TC (refer to Fig. 10), these differences are not systematic and reach about ± 30 mGal.





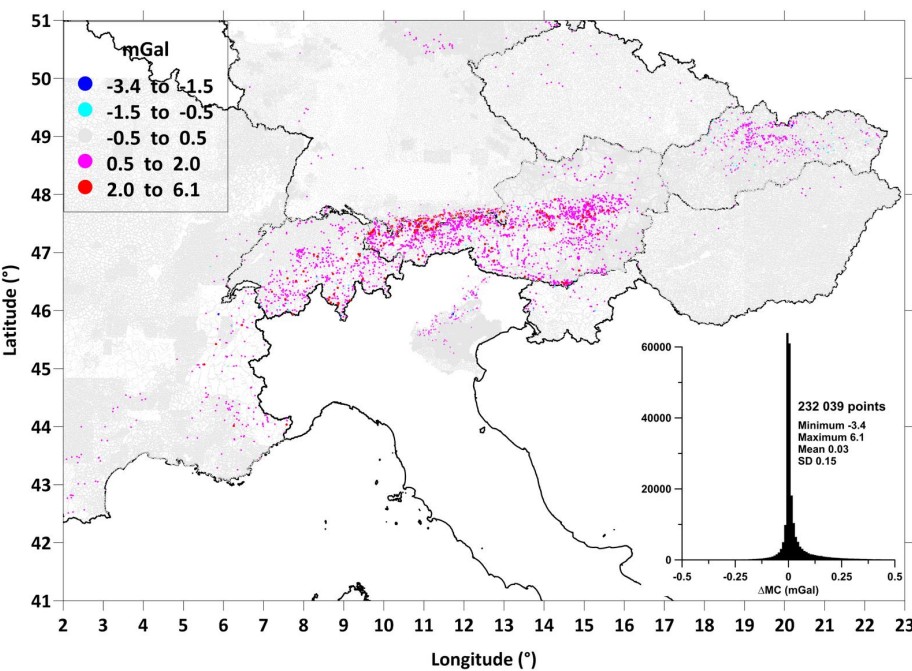


**Figure 13: Map of bathymetric corrections (up to the distance of 166.7 km, density 1640 kg m⁻³). Only non-zero values are shown on the map within 167 km of the sea. Shaded relief in the background shows the bathymetry of the seabed. The graph in the upper right corner shows the depth-dependence of bathymetric corrections. The red line represents their simple "Bouguer plate" approximation for comparison.**


## 4.4 Lake correction

Because the DEMs used in the MC calculation also include the volumes of water masses of Alpine lakes, these volumes are calculated with an incorrect density (2670 instead of 1000 kg m⁻³). We can eliminate this discrepancy by application of a lake correction. Steinhauser et al. (1990) point out that some Alpine lakes reach a depth of up to 300 m and due to easy accessibility

gravity stations are frequently located close to lake shores. An important prerequisite for a correct calculation is the availability of adequate models of lake bottoms. Depth models of Alpine lakes are available for four countries: Switzerland, Austria, Germany, and Slovenia.

For many large lakes in Switzerland bathymetric surveys have been carried out since 2007. The resolution of these models varies between 1 to 3 m. For all the other lakes which contain bathymetric contours in the topographic map 1 : 25 000, these

contours have been digitized and interpolated to grids of a resolution of 25 m.

In Slovenia there are two big Alpine lakes of glacial origin, located in the Julian Alps in the NW part of the country. For both lakes, high-resolution bathymetric data are available. Bathymetric surveys were performed in the years 2015-2017 (Harpha Sea, 2017). The maximum depths for Bohinj lake and Bled lake are 45 m and 30 m, respectively. The bathymetric grid size of 20 m was used to compute the alpine lake corrections for the new CBA.





No digital depth information was available for Austrian lakes. Therefore, shorelines and bathymetric contour lines have been digitized from topographic maps and interpolated to grids with 10 m spacing. All lakes (in total 36) exceeding either water volume of $25 \times 10^6$ m³ or maximum depth of 50 m have been handled in this way, including artificial reservoirs. The altitude of the lake level surfaces was derived from topographic maps too. Seasonal lake level variations cannot be ruled out; however, they are expected to be less than 1-2 m for natural lakes. The situation may be worse for reservoirs.

The depths data for lakes in the German parts of the Northern Alps model was digitized from topographical maps 1 : 50 000. The resolution is 25 m or 1 arcsec. Vertical heights are physical (normal) heights.

Mentioned models were combined with existing detailed DEMs, and the lake correction itself was calculated as the difference in gravitational effects of two topography models, one containing the level of the lakes and the other their bottom (Fig. 14). Calculated lake corrections (density 1670 kg m⁻³) for all countries with available lake models are in Fig. 15. The corrections

reach maximum values of about 5 mGal, especially on the lakesides with steep mountain flanks.

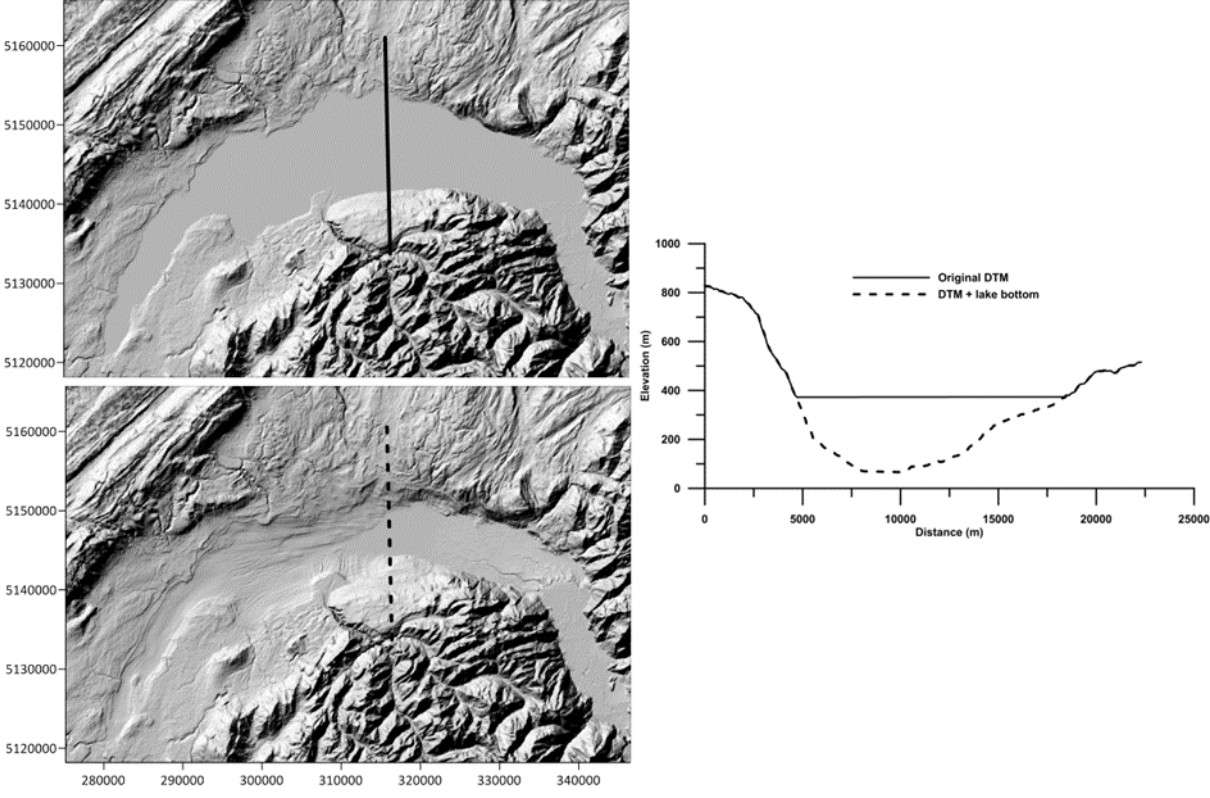

**Figure 14: Examples of topography models used to calculate lake corrections (here, Lake Geneva, Switzerland). Top shaded relief represents the original DEM (MERIT), the bottom one the combination of DEM and lake bottom. The graph on the right shows two profile lines crossing both models (North is to the right).**



Earth System
Science
Data

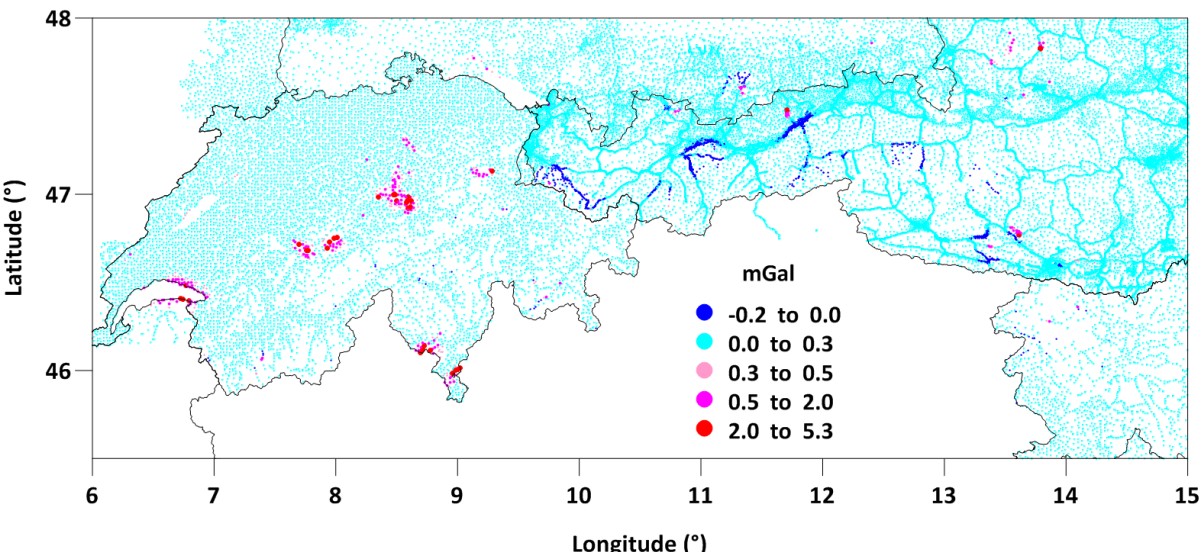

**Figure 15: Map of lake corrections (correction density is 1670 kg m⁻³). Small negative values occur in deep valleys with topography below the level of lakes (dark blue points).**

## 4.5 Digitization and reprocessing of the CBA map of the former SFR Yugoslavia

Although the peripheral SE part of the new Bouguer gravity map is not covered by terrestrial data which were available to the project, this area was filled by the digitization of the CBA map of the former SFR Yugoslavia at a scale of 1 : 500 000 (Bilibajkič et al., 1979). The CBA map (with a correction density of 2670 kg m⁻³) was published in 1972 and covers the whole area of the former SFR Yugoslavia. Its northern part was converted into an electronic form within the diploma thesis of Grand (2019). For the needs of the AlpArray project, a map was used especially for the territory of Serbia and Bosnia and Herzegovina. The gravity data of Slovenia and Croatia were also originally part of the Yugoslav gravity map (refer to Sect. 2 - Croatia). In contrast to the digitization for the AAGRG described here, the Slovenian and Croatian database contains new data.

The reprocessing included identification and correction of individual steps in the frame of CBA calculations to ensure a processing status which complies with that of the recalculated anomaly of the new AlpArray map. Specifically, normal gravity was corrected for the difference between the IGF 1967 and the Somigliana/GRS80 equations. Then the simple Free-Air correction was replaced by a more accurate approach, and the sphericity of the Earth was taken into account. However, this was neglected in cases where simple planar Bouguer corrections in the original data were used. For the last two corrections, the approximate heights at the digitization points generated from the model MERIT were used. Finally, atmospheric correction was calculated which was not considered in the original CBA. These reprocessing steps remained problematic, as the uniform procedure of their calculation was not used for the original CBA map and the original values were not published. Therefore,





given that MC/BC could not be recalculated and replaced by new values, we could expect more significant errors in the
transformed CBA. Fig. 16 shows a comparison of transformed CBA map with a map constructed from available data within
the project for Croatia. Fortunately, the differences between the maps are not significantly large, the standard deviation of
differences is about 1.8 mGal, with a low systematic difference (the mean value of the differences is less than 0.5 mGal). We
therefore assume that the replaced anomaly in the south-east part of the map (Serbia, Bosnia and Herzegovina) is of similar
quality than the main part.


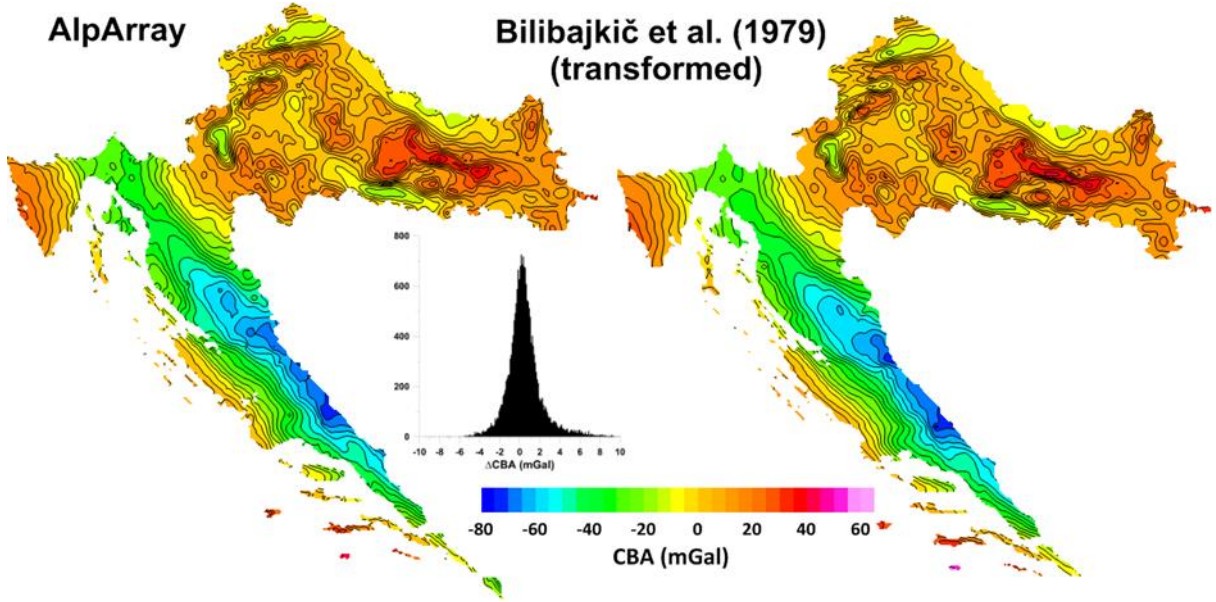

**Figure 16: Comparison of CBA maps (density 2670 kg m⁻³) for the area of Croatia. The map on the left is constructed from available data within the AlpArray project. The map on the right was obtained by transforming the digitized map of the former SFR Yugoslavia (Bilibajkič et al., 1979). The histogram in the middle shows the differences between the maps.**

**4.6 A short remark on future treatment of true atmosphere and distant relief effects**

As a challenge for the further development of the AlpArray CBA map, we also estimated the global effects of the true
atmosphere and distant relief.

Atmospheric correction is usually calculated based on a simple approximation according to Wenzel (1985). By the term true
atmosphere, we mean the model of the atmosphere derived from the effect of a spherical shell with radially dependent density
using the US standard atmosphere 1976 (Karcol, 2011) with an irregularly shaped bottom surface formed by the Earth's surface,
calculated globally (Mikuška et al., 2008). Difference between atmospheric correction calculated by both approaches for the

AlpArray region (calculated for selected database points) is shown in Fig. 17. The differences reach a maximum of about 0.16

mGal. As a function of height (approx. 0.04 mGal km$^{-1}$) it mainly depends on the topography and to a much lesser extent also

to the density model. Using a linear approximation instead of a time-consuming calculation at specific points would lead to

maximum errors of about 0.02 mGal. Note that in order to maintain the real situation regarding the distribution of atmospheric

masses, we used physical heights, not ellipsoidal.

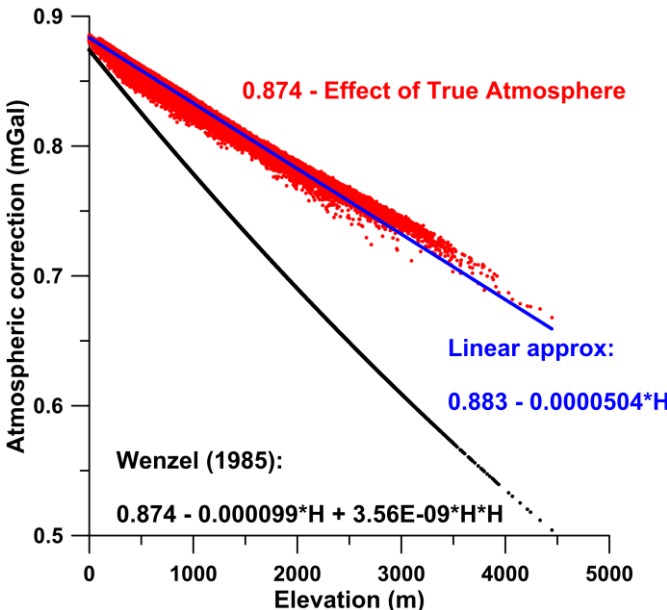

**Figure 17: Comparison of atmospheric correction at selected points covering the whole AlpArray area. The black dots represent the**
**atmospheric correction calculated by a simple approximation according to Wenzel (1985). The red dots show the calculation using**
**the effect of true atmosphere subtracted from the global constant value of 0.874 mGal  (Mikuška et al., 2008) and the blue line is its**
**linear approximation.**

Distant Relief Effect (DRE) represents the combined effect of topography and bathymetry beyond a standard distance of 166.7

km around the whole Earth (refer to Mikuška et al., 2006 for more detailed information). Fig. 18 shows this effect calculated

at selected points in the AlpArray study area. The calculation was made in the classical concept of physical heights. The

calculation for ellipsoidal heights would differ slightly (in quantitative terms), but the basic features would be retained as

presented. The inclusion of this effect in the CBA is a task for future studies. DRE is dominated mainly by long-wavelength

trends, superimposing also high-frequency patterns in particular in mountainous regions due to its dependence on height.

Because terrain masses are largely compensated by isostatic compensation, distant compensating mass distribution should be

considered as well (e.g., Szwillus et al., 2016), either by applying isostatic concepts or by relying on global crust-mantle

boundary models. However, these additional considerations are beyond the main objective of this publication.


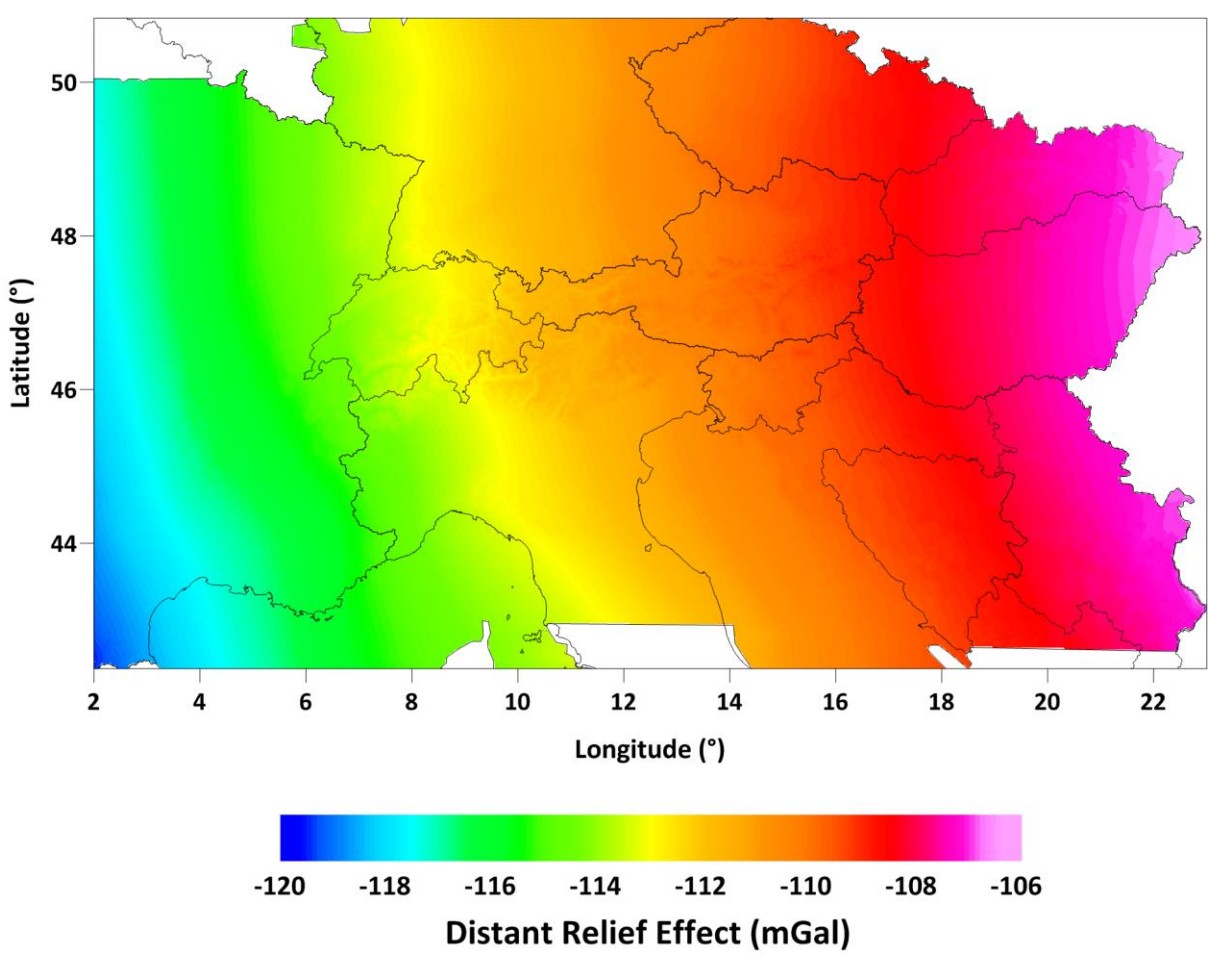

**Figure 18: The summary effect of topography and bathymetry (densities 2670 kg m⁻³ and -1640 kg m⁻³, respectively) from 166.7 km**
**around the whole Earth.**

## 5 The new homogenized gravity maps for the Alps

### 5.1 Interpolation and reference height of interpolated Bouguer anomalies

AlpArray gravity data have different levels of confidentiality. In some cases, only interpolated grids are available. Therefore, well defined interpolation procedures are required. Interpolating scattered gravity data onto regular grids is commonly done in

2D, ignoring the fact that original data is acquired at different elevations rather than at a constant level. More exact solutions





would be achieved by solving a proper boundary value problem. However, those methods are very time consuming, and avoiding mathematical artefacts due to limitation of data in terms of spatial extent and resolution is not trivial at all. Hence, the AAGRG decided to provide grids based on 2D interpolation first.

For assessing the 2D interpolation error in rugged terrain, two synthetic gravity data sets have been created based on two
different kinds of source representation: a polyhedron model (method by Götze and Lahmeyer, 1988) and an equivalent source model (EQS) determined by the method of Cordell (1971). The model response has been calculated at the scattered positions of a subset of Austrian gravity data as well as at the grid nodes with 1 km spacing. The synthetic data sets almost keep the wave-length content of real world data. The elevation at the grid nodes was interpolated by 2D-Kriging based on the scattered data information.

In the case of the polyhedron model, the differences between exact 3D-prediction and 2D interpolation do not exceed the range of 1-2 mGal. Only in small, isolated areas the errors are larger than 5 mGal. The same holds for the equivalent source representation where the errors are in the range of ± 1 mGal and exceed ± 2 mGal only at a few spots (Fig. 19).


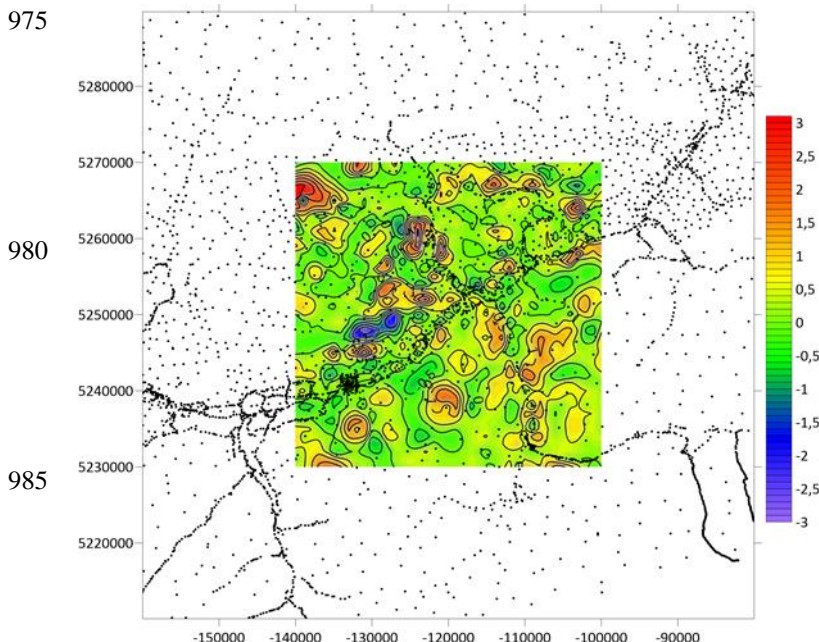



**Figure 19: Interpolation error estimate (gravity difference between gravity fields predicted by the EQS model and by 2D interpolation, contour interval 0.1 mGal, axis coordinates in [m] (Gauss-Krüger projection, M31).**





In large scale 3D modelling, 3D models rarely match the data better than the errors estimated in the scenarios tested above. Therefore, 2D interpolation seems to be justified even if it is not exact from a theoretical point of view. In local-scale
interpretation, the situation may be different.

However, another problem arises when using interpolated grids. Modelers need to know the elevation at which interpolated Bouguer or Free air anomalies refer to.

Assuming the interpolation operator to be linear, Bouguer anomaly (*BA)* and Free air anomaly (*FA)* interpolated at each grid node $(x_i, y_j)$ read as

$$\text{BA}_{\text{int}}(x_i, y_j) = g_{\text{int}}(x_i, y_j) - \gamma_{\text{int}}(x_i, y_j) - \text{MC}_{\text{int}}(x_i, y_j) \tag{7}$$

$$\text{FA}_{\text{int}}(x_i, y_j) = g_{\text{int}}(x_i, y_j) - \gamma_{\text{int}}(x_i, y_j) \tag{8}$$

where the suffix "int" denotes interpolated quantities and MC is the gravitational effect of surplus and deficit mass with respect to the reference ellipsoid. By transforming Eq. (7) and using Eq. (8) we get

$$\text{FA}_{\text{int}}(x_i, y_j) = \text{BA}_{\text{int}}(x_i, y_j) + \text{MC}_{\text{int}}(x_i, y_j) \tag{9}$$

Assuming the Bouguer anomaly to be a sufficiently smooth function of horizontal coordinates, true gravity at the position $(x_i, y_j)$ of a grid node and at the true elevation $h_{\text{topo}}(x_i, y_j)$ can be approximated by

$$g(x_i, y_j, h_{\text{topo}}) \approx g_{\text{rec}}(x_i, y_j, h_{\text{topo}}) = \text{BA}_{\text{int}}(x_i, y_j) + \gamma(x_i, y_j, h_{\text{topo}}) + \text{MC}(x_i, y_j, h_{\text{topo}}) \tag{10}$$

where the suffix "rec" denotes approximated (reconstructed) quantities.

The Bouguer anomaly at grid node $(x_i, y_j)$ and at true elevation $h_{\text{topo}}(x_i, y_j)$ is

$$\text{BA}(x_i, y_j, h_{\text{topo}}) = g(x_i, y_j, h_{\text{topo}}) - \gamma(x_i, y_j, h_{\text{topo}}) - \text{MC}(x_i, y_j, h_{\text{topo}}) \tag{11}$$

Approximating $g(x_i, y_j, h_{\text{topo}})$ by Eq. (10) and inserting into Eq. (11) results to

$$\text{BA}(x_i, y_j, h_{\text{topo}}) \approx \text{BA}_{\text{int}}(x_i, y_j) + \gamma(x_i, y_j, h_{\text{topo}}) + \text{MC}(x_i, y_j, h_{\text{topo}}) - \gamma(x_i, y_j, h_{\text{topo}}) - \text{MC}(x_i, y_j, h_{\text{topo}})$$

or

$$\text{BA}(x_i, y_j, h_{\text{topo}}) \approx \text{BA}_{\text{int}}(x_i, y_j) \tag{12}$$

However, this approach neglects that the Bouguer anomaly is the gravity effect of all sources at the true location of a station and therefore depends on the station heights as well. We would get the same result as in Eq. (12) for any arbitrary elevation $h$ used in Eqs. (10) to (12), also for $h_{\text{int}}$. Hence, we can interpret the interpolated Bouguer anomaly as being valid at the true elevation $h_{\text{topo}}(x_i, y_j)$ of a grid node $(x_i, y_j)$ but also at elevation $h_{\text{int}}$. Because interpolation is always associated with smoothing we can argue that the best location for referencing the Bouguer anomaly is $h_{\text{int}}$. If modelers use true elevations for the grid
nodes, then models based on polyhedron approaches suffer from an aliasing problem, because the topography is not well represented by the grid. A smoothed (interpolated) topography would work better because interpolation includes a kind of filtering.





Particularly in rugged terrain, FA and MC are not smooth functions of horizontal coordinates. Therefore applying Eq. (9) is rather questionable. Instead, the Free air anomaly at a grid node $(x_i, y_j)$ and at true elevation $h_{topo}(x_i, y_j)$ can be better

approximated by

$$FA(x_i, y_j, h_{topo}) \approx FA_{rec}(x_i, y_j, h_{topo}) = g_{rec}(x_i, y_j, h_{topo}) - \gamma(x_i, y_j, h_{topo}) = BA_{int}(x_i, y_j) + MC(x_i, y_j, h_{topo}) \qquad (13)$$

Inserting Eq. (7) into Eq. (13) results to

$$FA(x_i, y_j, h_{topo}) \approx g_{int}(x_i, y_j) - \gamma_{int}(x_i, y_j) - MC_{int}(x_i, y_j) + MC(x_i, y_j, h_{topo})$$

or with Eq. (8)

$$FA(x_i, y_j, h_{topo}) \approx FA_{int}(x_i, y_j) - MC_{int}(x_i, y_j) + MC(x_i, y_j, h_{topo}) \qquad (14)$$

The free air anomaly at the true elevation $h_{topo}(x_i, y_j)$ of a grid node $(x_i, y_j)$ can be reconstructed either by Eq. (13) or Eq. (14).

However, also in this case we have to keep in mind that we actually do not overcome the problem of the height dependence of Bouguer anomalies. When we use $h_{int}$ instead of $h_{topo}$, Eq. (13) and Eq. (14) hold accordingly.

Note that we implicitly also included bathymetry in the MC-term appearing in Eqs. (7) to (14). Regarding the Bouguer anomaly

$BA_\rho$ calculated with density $\rho$ differing from density $\rho_0$ used in the mass correction term MC in Eqs. (7) to (14) we have to separate liquid from solid parts, which leads to the following equation:

$$BA_\rho(x_i, y_j) = BA_{int}(x_i, y_j) + \left(1 - \frac{\rho}{\rho_0}\right) \delta g_M(x_i, y_j) - \left(1 - \frac{\rho - \rho_{oc}}{\rho_0 - \rho_{oc}}\right) \delta g_B(x_i, y_j) \qquad (15)$$

where $\rho_{oc}$ is the density of ocean water (1030 kg m$^{-3}$).

Eq. (15) neglects the small density difference between lake and ocean water. However, this leads to only very small errors in

the order of a few % of the lake correction for reasonable crustal densities.

**To conclude:**

> **In addition to the methodological procedures just described, we will now discuss another problem related to the gridding of our data base. In case of the AAGRG compilation, interpolation of original and gridded data has been done by an iterative procedure:**
>
> **(a) Data providers, who were not allowed to release original information, created gridded data relying initially on their own scattered data and keeping only the nodes inside their own territory, on a grid the AAGRG defined in common for the whole area.**
>
> **(b) After merging all data sets from AAGRG members one common grid is interpolated.**
>
> **(c) In the next step grid nodes of the neighbouring countries were merged with the provider´s original data set, and a new data grid is interpolated.**
>
> **(d) This iterative procedure is continued until the variation of interpolated grid data close to the borders is well below an error threshold defined by ±5 mGal.**





## 5.2 Filling data gaps using Global Geopotential Models (GGM)

We have focused on commonly used Global Geopotential Models (GGM) up to the degree/order of 2190, mainly EIGEN-6C4 elaborated jointly by GFZ Potsdam and GRGS Toulouse (Förste et al., 2014) and EGM2008 (Pavlis et al., 2012). Both models are created by the combination of satellite and terrestrial gravity data. The spatial resolution of these models is roughly about 10 km.

The GGM models are usually used in connection with the so-called Residual Terrain Modeling (RTM) technique, which greatly improves gravity values calculated from GGM on the Earth surface. The RTM technique accounts for the difference between the gravitational effect of the real terrain masses represented by high-resolution DEMs, and smoothed mean elevation surface represented e.g., by the DTM2006 model (Pavlis et al., 2007). However, since the effect of the detailed DEM would be subtracted retrospectively in the Bouguer anomaly calculation, it means that, in order to obtain BA, we only need to subtract the gravity effect of the DTM2006 ($\delta g_{DTM2006}(\lambda, \varphi, h_E)$) directly from the Free-Air anomaly calculated from GGM-derived gravity by the standard procedure of Eq. (16). Compared to Eq. (1), Eq. (16) lacks the term for the atmospheric correction because it is already included in the GGM.:

$$BA_{GGM}(\lambda, \varphi, h_E) = g_{GGM}(\lambda, \varphi, h_E) - \gamma(\varphi, h_E) - \delta g_{DTM2006}(\lambda, \varphi, h_E), \quad (16)$$

where $g_{GGM}$ is the gravity calculated from a particular GGM at the Earth surface (to be directly comparable with the terrestrial data) at elevations derived from MERIT model, $\gamma$ is the normal gravity, and $\delta g_{DTM2006}$ is the gravitational (terrain and bathymetry) effect related to the model DTM2006 (of the corresponding degree of 2190) up to the distance of 166.7 km. The DTM2006 model was selected due to its close relationship with the creation of the model EGM2008. This model was originally compiled in a grid of $30'' \times 30''$. For the purposes of our calculations, the model was transformed and resampled into a format corresponding to the calculation of the standard mass/bathymetric correction using Toposk.

We calculated the gravity values $g_{GGM}$ using the software GrafLab (Bucha and Janák, 2013) using the maximum degree of spherical harmonic coefficients for a specific GGM. Calculations were performed in GRS80 ellipsoidal coordinates.

The Fig. 20 shows a comparison of BA map derived from terrestrial data with the map derived from the EIGEN-6C4 model (calculation points were made on a $2 \times 2$ km grid) in the area covered by terrestrial data. The maximum differences between grids are at the level of tens of mGal (RMS error is about 4 mGal), but without any systematic error. It follows that the GGM-derived map can be used to fill in gaps (marginal parts) in the terrestrial data.

GGM data points located in gaps of the original gravity points were separated by the shortest distance criteria of 15 km using a standard database search query in QGIS. The 15 km criterion was chosen as a compromise between covering GGM data close enough to the vicinity of the terrestrial data (Fig. 27), but at the same time not to fill too small gaps between them, which could lead to local artificial anomalies.


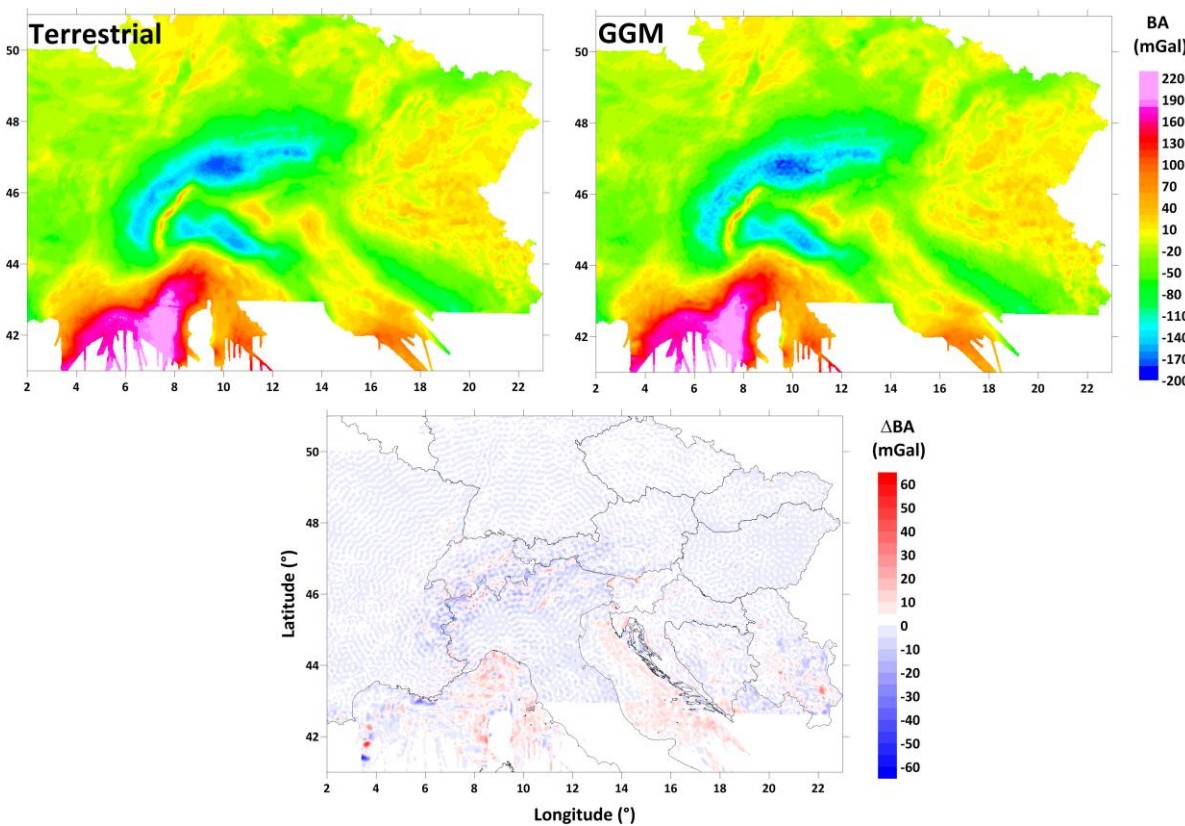

**Figure 20: Comparison of Bouguer anomaly maps (correction density 2670 kg m$^{-3}$) derived from terrestrial data (upper left) and**
**GGM model EIGEN-6C4 (upper right). The bottom map shows the difference between the two.**

### 5.3 Brief interpretation of Bouguer anomaly map

We here present a short overview of the features of the new Bouguer anomaly map (Fig. 21). A high resolution 600 dpi plot
of the map is available in the supplement. The most prominent feature of the complete Bouguer anomaly (CBA) is the Alpine
gravity low (AGL), which is characterized by gravity values ranging from -100 to -170 mGal. The AGL corresponds with the
Alpine mountain chain and is explained by the isostatic crustal thickening,
as demonstrated by the good anticorrelation with topography (Braitenberg et al., 2013; Pivetta and Braitenberg, 2020) and the
isostatic compensation and gravity forward models (e.g., Ebbing et al., 2006; Braitenberg et al., 2002). It could be divided into
local gravity lows that correlate with the Western, Central and Eastern Alps. Among all of them the Central Alps (the
easternmost part of Switzerland) are accompanied by the highest amplitude -170 mGal.
A second prominent low is the Po Basin gravity low (PoBGL). The gravity values here range from about -80 to -140 mGal.
The PoBGL continues in the SE direction to the Central Apennine gravity low (CAGL), whose amplitude (-40 mGal) is





significantly smaller in comparison with the Northern Apennines gravity low. In the southeasternmost part of the Central Apennines the CAGL thins out gradually.

A significant anomaly feature represented by very narrow local gravity high can be clearly recognized between the Western Alps and the Po Basin. This anomaly is well known as the Ivrea gravity high (IGH). It is characterized by maximum values of +40 mGal, caused by dense, lower crustal and mantle rocks that are exposed and in the near subsurface, and which are planned to be drilled in the forthcoming DIVE project (Pistone et al. 2017; http://dive.icdp-online.org/). It is important to note that its relative amplitude against the gravity lows in the Western Alps and the Po Basin reaches up to 160 mGal. It is the highest

horizontal gravity gradient in the study region.

To the north-east of the Po Basin, we can observe the Verona-Vicenza gravity high (VVGH), which has been recently modelled as being generated by increased density crustal intrusions related to the Venetian magmatic province (Tadiello and Braitenberg, 2021; Ebbing et al., 2002). The Venetian-Friuli Plain gravity low (VFGL) is located in eastern Italy, which is presumably caused by low density sedimentary infill, as also the gravity low in the Po Basin (Braitenberg et al., 2013).

A prominent gravity high is the Mediterranean gravity high (MGHi). This regional scale anomaly has its maximum over the Corso-Ligurian Basin, the Corso-Ligurian gravity high (CLGH). It is characterized by maximum values of +200 mGal. The regional MGHi also includes the Tyrrhenian gravity high (TGH). The study covers only the northern part Gravity values do not exceed +140 mGal. The Corso-Ligurian gravity high and the Tyrrhenian gravity high are separated from the relative Corsica-Sardinia gravity low (CSGL). The values vary from +20 to +60 mGal.

The Adriatic Sea region is largely characterized by a positive gravity field, in which the south Adriatic gravity high (SAGH) dominates with values from +20 to +100 mGal. Its maximum is located over the Gargano promontory. In the north-western part of the Adriatic Sea, negative gravity values up to -80 mGal are observed, which belong to the easternmost part of the Po basin gravity low. West of the Istrian peninsula the centre of residual Istria gravity high (IGH) is present, with maximum values of +30 mGal.

In the Eastern Alps, the AGL splits towards the east into two branches of less pronounced gravity lows: The Western Carpathian gravity low (WCGL) and the Dinaric gravity low (DGL). In the Western Carpathians, the values vary from 0 to -60 mGal, while the Dinarides range 0 to -120 mGal. The lower amplitude of the gravity field of both the WCGL and the DGL in comparison with the AGL most likely reflects a weaker continental collision resulting in thinner crust under the Carpathians and Dinarides. In the Adriatic region we can also recognize the Merdita gravity high (MeGH) and the pre-Adriatic gravity low

(ADGL).

The Pannonian Basin extending between the Western Carpathians and the Dinarides is accompanied by relative regional gravity high (PBGH) whose values range in a narrow interval from -10 to +20 mGal. The PBGH consists of several local positive [the Transdanubian gravity high (TDGH), the Papuk gravity high (PGH), the Mecsek gravity high (MsGH), the Fruška Gora gravity high (FGGH)] and negative anomalies [the Danube Basin (DBGL) and the Makó-Békés Basin (MBGL)]. The

gravity effect of the Apuseni Mts. is negative (maximum -80 mGal).



The rest of the study area extending north of the MGHi, AGL and WCGL is accompanied by an indistinct, yet variable gravity field with the values varying generally from -80 to +40 mGal. Based on the analysis of the gravity field in this area, we recognize the following anomalies: the Pyrenean gravity low (PGL), the Massif Central gravity low (MCGL), the Paris Basin gravity low (PBGL), the Upper Rhine graben gravity low (URGL) and the Rhône-Bresse Graben gravity high (RBGH), the

Black Forest gravity high (BFGH) and the Vosgesian gravity high (VGH).

The gravity field of the Bohemian Massif can be divided into several sub-parallel positive (up to +20 mGal) and negative (0 to -60 mGal) belts with predominantly NE-SW orientation: the Krušné hory (Erzgebirge)-Krkonoše gravity low (KKGL), the Teplá-Barrandian-Labe gravity high (TBLGH), the Moldanubian gravity low (MGL), the Orlice-Opole gravity low (OOGL), the Moravo-Silesian gravity high (MSGH), the Upper Silesian gravity high (USGH) and the Sudetes gravity high (SGH).

The gravity field over the Franconian Platform area north of the Molasse Basin is quite variable and values range from -40 mGal to +15 mGal. The eastern part of the Franconian Platform is characterized predominantly by negative, while the western part by positive values.

The Rhenish Massif is distinctly asymmetric, positive (up to approx. +20 mGal) over the eastern massif and negative (to approx. -20 mGal) over the western massif. The Ardennes are accompanied by the gravity low of -20 mGal. The Brabant

Massif is manifested by a gravity high with amplitude +20 mGal.

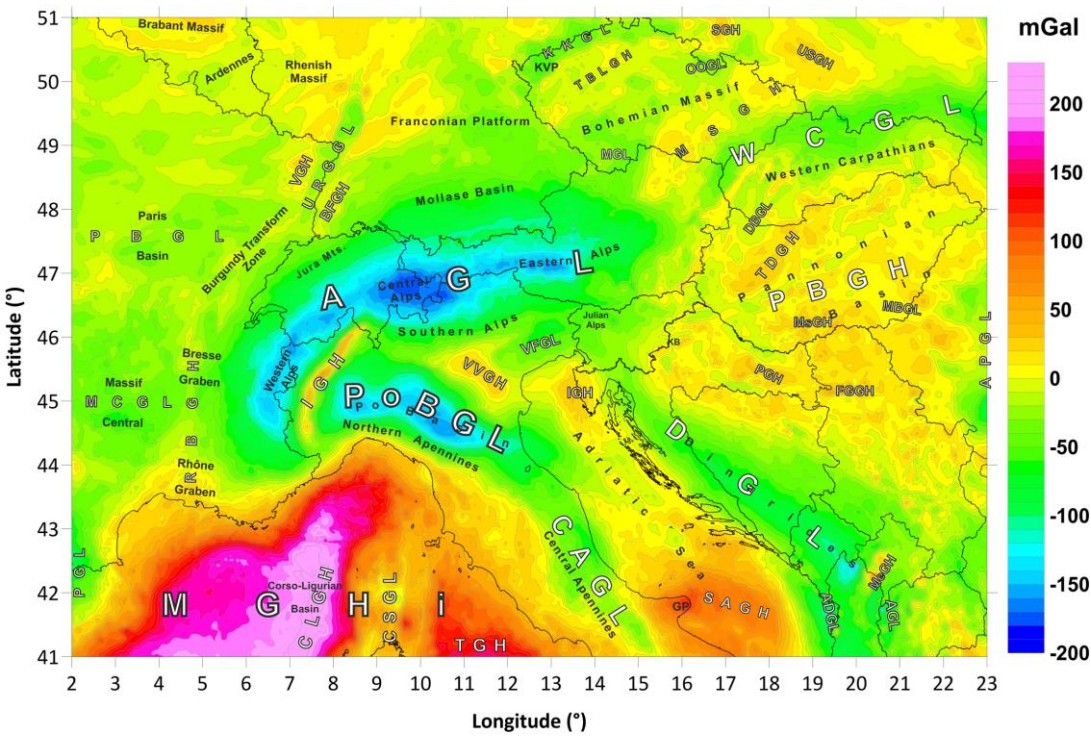

**Figure 21: New Pan-Alpine Bouguer gravity anomaly map. The first order dominant regional gravity anomalies: AGL - Alpine gravity low, PoBGL - Po Basin gravity low, CAGL - the Central Apennine gravity low, IGH - Ivrea gravity high, VVGH - Verona-**



**Vicenza gravity high, VFGL - Venetian-Friuli Plain gravity low. The second dominant regional gravity anomaly: MGHi - Mediterranean gravity high, CLGH - Corso-Ligurian gravity high, TGH - Tyrrhenian gravity high, CSGL - Corsica-Sardinia gravity low, SAGH - south Adriatic gravity high, IGH - Istria gravity high (IGH), WCGL - Western Carpathian gravity low, DGL - Dinaric gravity low, MeGH - Merdita gravity high, ADGL - pre-Adriatic depression, PBGH - Pannonian Basin gravity high, TDGH - Transdanubian gravity high, PGH - Papuk gravity high, MsGH - Mecsek gravity high, FGGH - Fruška Gora gravity high, DBGL**
**- Danube Basin gravity low, MBGL - Makó-Békés Basin gravity low, APGL - Apuseni gravity low. The rest of the study area: PGL - Pyrenean gravity low, MCGL - Massif Central gravity low, PBGL - Paris Basin gravity low, URGGL - Upper Rhine graben, RBGH - Rhône-Bresse Graben gravity high, BFGH - Black Forest gravity high, VGH - Vosgesian gravity high, KKGL - Krušné hory (Erzgebirge)-Krkonoše gravity low, TBLGH - Tepla-Barrandian-Labe gravity high, MGL - Moldanubic gravity low, OOGL - Orlice-Opole gravity low, MSGH - Moravo-Silesian gravity high, USGH - Upper Silesian gravity high, SGH - Sudetes gravity high,**
**KB - Krško Basin.**

## 6 Uncertainties of data and map

The newly compiled gravity database of the Alps and their surroundings is based on decades of data collection and processing experience of the AAGRG members. The national gravity data, which were recompiled here under new, modern geophysical-geodetic aspects (Sects. 2 and 4), were collected with rather different instruments at different times over the last 70 years and
processed with extremely different processing methods. At the end of the data processing, we therefore asked ourselves for what purposes it can be used and how accurate the new map actually is. The first question can be answered relatively easily: with medium to large scale modelling of the Alpine lithosphere and/or the Alpine Earth crust, as realized in the AlpArray initiative, there should be no problems with the final accuracy of database: these errors are small compared to the uncertainties that result from modelling and simulation. The second question about accuracy (uncertainty), which is caused using extremely
different data sets, is much more difficult to answer because in practice for all participating countries there are no exploitable metadata available for the national gravity databases.

As desirable as it would have been for the submitted pan-Alpine gravity maps to present "uncertainty maps" at the same scale, this project is hindered due to the complexity of the task and the lack of information on errors and accuracies in the field
campaigns and data processing of the individual countries. However, in order to obtain an estimate of the uncertainty, we have tried in the following Sect. to list various aspects of error analysis by way of examples. It must be reserved for a later publication to present a numerical-statistical analysis of the map (e.g., with the time consuming "Sequential Gaussian Simulation", e.g., Shahrokh et al., 2015) or statistical evaluation against the GOCE gravity observations, that have lower spatial resolution, but homogeneous error (Bomfim et al., 2013).

**Testing at independent gravity points - example from Slovakia**

In Fig. 22 we show a test calculation that demonstrates the differences between the fields of the interpolated CBA and point stations in Slovakia. These "test data" have not been taken into account for the interpolation of the Slovakian gravity grid - thus they represent an independent test of the map quality. First, it should be noted that no deviations are greater than ±5 mGal. The mean is 6 μGal and the standard deviation is 0.88 mGal. This is an ideal example for visualizing "mapping errors" which

are expected in case of a dense and widely homogeneous data coverage. However, in areas of less dense and less homogeneous

coverage like along the Alpine crests or in the offshore areas the number of errors increases.

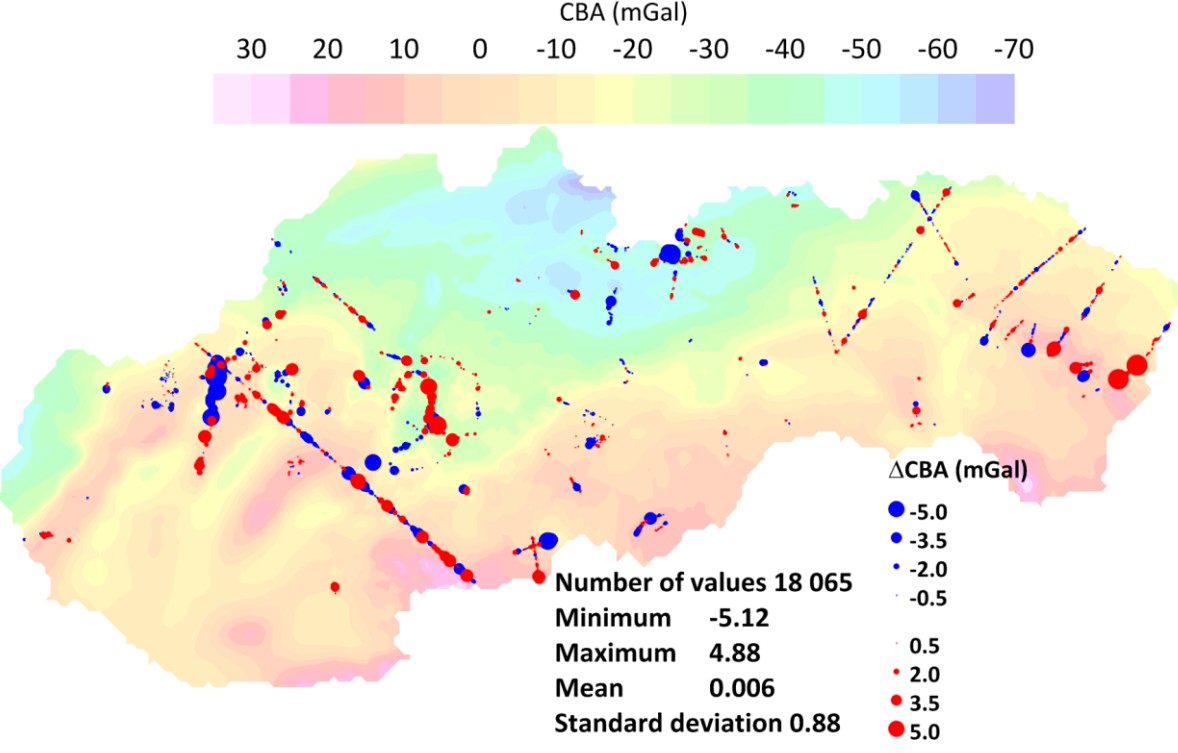

**Figure 22: Differences between the CBA grid and independent gravity points (not used for the Slovakian part of the gravity grid**
**compilation). It was calculated by SURFER´s simple grid-residual procedure and showed that no gravity differences were greater**
**than ± 5 mGal.**

### 6.1 Possible sources of errors

The sources of errors in gravimetric measurements are manifold and result directly from the definition of the Bouguer anomaly
and the processing of associated reduction and correction terms (Sect. 3, Eq. (1)). Instrumental readings in gravimetry depend

on the instrument drift and the accuracy of the scale values and are of course dependent on the external conditions in the field.

In addition, there is a correction of the Earth tides and the air pressure. The localization of the station with longitude, latitude

and altitude as well as its geographical context (e.g., measured along profiles, areal measurements, located in valleys with big

sedimentary filling etc.) is also subject to errors. The density of the station distribution (Fig. 1) certainly has a great influence
on the accuracy of the resulting maps. This is, however, good enough for the above-mentioned modelling of the lithosphere -

very small-scale modelling on a km-scale is excluded.

Even the indication of the positional accuracy of the gravity stations and the DEMs used pose great problems and most of the information is not available in digital formats. The same is true for the above-mentioned field instruments and procedures used, which have improved often over the last 70 years, and of course for the processing techniques, which started with manual-graphic methods and still allow digitized processing from field measurements to 3D interpretation (among many others: Cattin et al., 2014; Schmidt, pers. communication).

One more word about the different results with different methods. In Sect. 4.1. we reported test investigations which led to the selection of the software for the calculation of the MC (Fig. 8). A comparison of the standard deviations (1.95 mGal for the software Tritop and 0.39 mGal for Toposk) also gives an indication of the achieved accuracy of the database - even if this can only be a partial aspect.

Two other sources of error deserve a closer look: in Sect. 6.2 we will discuss errors that occur when calculating the mass correction with different correction densities. Notes on the accuracy of the anomalies due to a 2D (on the map projection plane) and a 3D interpolation to be demanded have already been given in Sect. 5.2. Based on national investigations in the area of Austria, indications of the achieved numerical accuracy of the Bouguer anomalies are then given in Sect. 6.3. Finally, in Sect. 6.4 the results of an error statistic based on cross validations (CV) is given for the entire database.

*However, it should not be forgotten that CV is a purely statistical measure and in minor amounts considers point data quality which indicates that we cannot directly represent the quality of the newly compiled gravity fields from the CV.*

CV works well with dense station coverage; only then we can exclude large local anomalies, for example due to geological causes. The less dense the coverage is, the less we can exclude the presence of local anomalies. Note, that these local anomalies can easily be produced by selecting improper MC density, for example, in a station setting covering a valley and adjacent mountain flanks where densities differ from the assumed MC density remarkably.

## 6.2     Errors in the calculation of mass corrections (MC)

The DEM used has a significant influence on the result. For example, it could be shown that the difference in the MC calculations using the LIDAR DEM and the MERIT DEM (Sect. 4.2) yielded values of ± 5 mGal.
Another question is whether MC errors related to the density uncertainty should be considered. Actually, the BA calculated even with any constant density has exact physical meaning as gravitational effect (i.e., vertical component of the gravitational vector) of sources differing w.r.t. density from the density of the reference ellipsoid inside the ellipsoid and from the assumed MC density outside the reference ellipsoid (e.g., Meurers, 2017). This interpretation makes clear how models must be formulated in 3D-modeling and interpretation. Generally, there are three options:

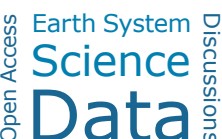

A: If constant density is used for MC, then 3D models have to describe the density contrast w.r.t. the reference ellipsoid (within the ellipsoid) and the chosen MC density. Of course, models must have the same spatial resolution as the DEM used for MC. Because this is practically impossible, a smoothed topography will be used in practice to keep the model parameter manageable.

B: If 2D density models are applied for MC, then these models have to be used as reference within the topography domain. Then, the starting model gets probably much more complex, while the model geometry needs to have the same spatial resolution as in case A. So, for modelers, 2D density models are no progress over option A.

C: If and only if true 3D density is used for MC, modelling would get „simple ", because then the upper surface of the model space gets smooth and coincides with the reference ellipsoid: no sources outside the ellipsoid.

However, option C can never be implemented in practice. If we would regard incorrect MC density as an error source, these errors can be as high as 700 kg m$^{-3}$ (e.g., in valleys). Then, the MC-error results from multiplying the density errors by the MC calculated with unit density and is likely of the order 30-50 mGal or higher. However, it would not make sense for modelers to take these numbers as adjustment threshold, because this is about 10-20% of the BA of the Alps. Therefore, it is advisable to keep the exact definition mentioned above in mind, and incorrectly chosen MC density does not lead to errors at all.


### 6.3    Mapping errors in selected areas of the map

As already discussed in Sect. 5.1, using 2D interpolation procedures in the map layer are not exact, but the errors are negligible in large scale interpretation. Here, we use two approaches for assessing the interpolation error: interpolation residuals and cross validation residuals. Interpolation residuals depend on the mathematical representation of the interpolation grid. We use the
bilinear interpolation method for calculating the residuals at points that do not coincide with grid nodes. Interpolation residuals describe how exact the scattered data are represented by the interpolation surface. Cross validation residuals are calculated by removing one observed station from the data set and using all remaining data to interpolate a value at its location. This procedure is repeated for all the other stations of the data set. Both methods reflect gross data errors if present. However, large residuals do not indicate data errors necessarily but hint to a possible sampling problem if a true local anomaly is not
sufficiently supported by the station coverage in the surrounding area. In the following, residuals are defined as interpolated minus observed value.

**Example Austria**

The interpolation residuals of the Austrian data set range between about -8 and +8 mGal, the cross validation residuals between -14 mGal and +10 mGal. Standard deviations are well below 1 mGal (Table 7).



|  | Interpolation residual | Cross Validation residual |
|---|---|---|
| Number of values | 50 492 | 51 464 |
| Minimum | -8.24 | -14.13 |
| Maximum | 7.66 | 9.94 |
| Mean | 0.11 | -0.03 |
| Variance | 0.77 | 0.81 |
| Standard deviation | 0.88 | 0.81 |


**Table 7: Residual statistics for the Austrian data set. Units in [mGal].**

For discussing the sampling problem, Fig. 23 shows the interpolation residuals (left) and the cross validation residuals (right) within a smaller section of the Enns valley in Austria. Background colours display the topography, contour lines show the

Bouguer anomaly interpolated to a high resolution grid with spacing of 0.002 65° in longitude and 0.001 73° in latitude corresponding to a grid spacing of about 200 m. The local negative BA reflects the gravitational effect of the low density sediment filling of the Enns valley. Coloured dots show the residuals as class scatter plot with respect to the AlpArray grid with about 2000 m spacing. The interpolation residuals range to about 6 mGal along the valley axis, while they are reduced to less than 1 mGal if calculated with respect to the high resolution grid. Large cross validation residuals are observed at these

stations as well. Given the spacing of 2000 m of the AlpArray grid, the interpolation algorithm does not capture the local anomaly. In this case, the interpolation residuals do not indicate BA errors but reflect the smoothing effect of the coarse AlpArray grid interpolation as it was already mentioned in the introduction to Sect. 6.

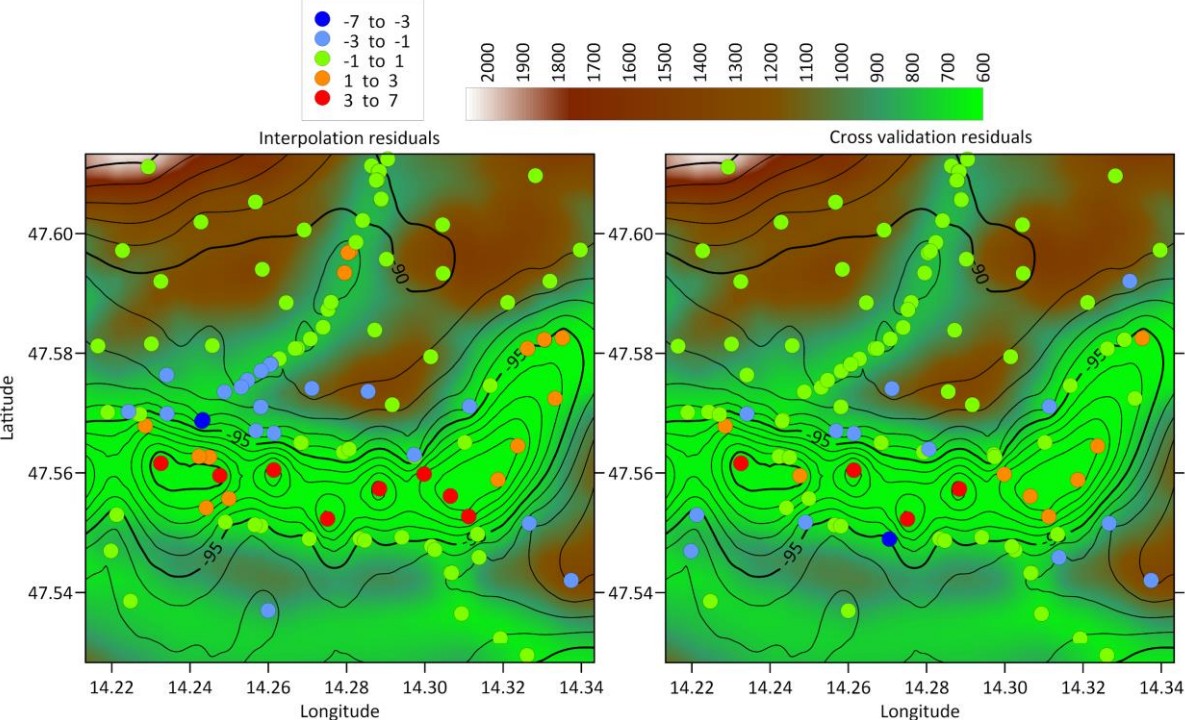

**Figure 23: Interpolation and cross validation residuals of a subset within a small section of the Enns valley in Austria. Background:**
**Topography, contour lines: CBA anomaly [mGal] interpolated using a high resolution grid (about 200 m spacing), coloured dots: residuals (left: interpolation, right: cross validation) at the scattered data points with respect to the AlpArray CBA grid (2000 m spacing). Residuals in [mGal], height in [m].**

## 6.4 Cross validation error for the entire database

As mentioned in the previous subsection both interpolation residuals and cross validation methods provide some picture of data quality. At the same time, these methods can be used as a criterion for excluding gross errors from individual databases. Both methods give qualitatively similar results (see Fig. 23), with cross validation giving quantitatively more significant residuals. Since in the case of cross validation residuals (unlike Interpolation residuals) it is possible to exchange data between grid providers in order to comply with the conditions of confidentiality of the original data, we show in Fig. 24 a complete

map of Cross validation residuals for the whole area. While the standard deviation of these residuals is well below 1 mGal (comparable to Table 7), the extreme values reach tens of mGal (about 650 points exceed 10 mGal, 16 points exceed 20 mGal). An extreme point with a residual higher than 60 mGal creates a characteristic bull-eye anomaly in the CBA map (Fig. 25). We consider similar points with extreme residuals to be erroneous and it is therefore necessary to exclude them from the database before compiling the final CBA map. Therefore, it is necessary to choose a reasonable criterion considering the analysis of

errors as well as the problem of inhomogeneous coverage of the territory by the data described in the previous



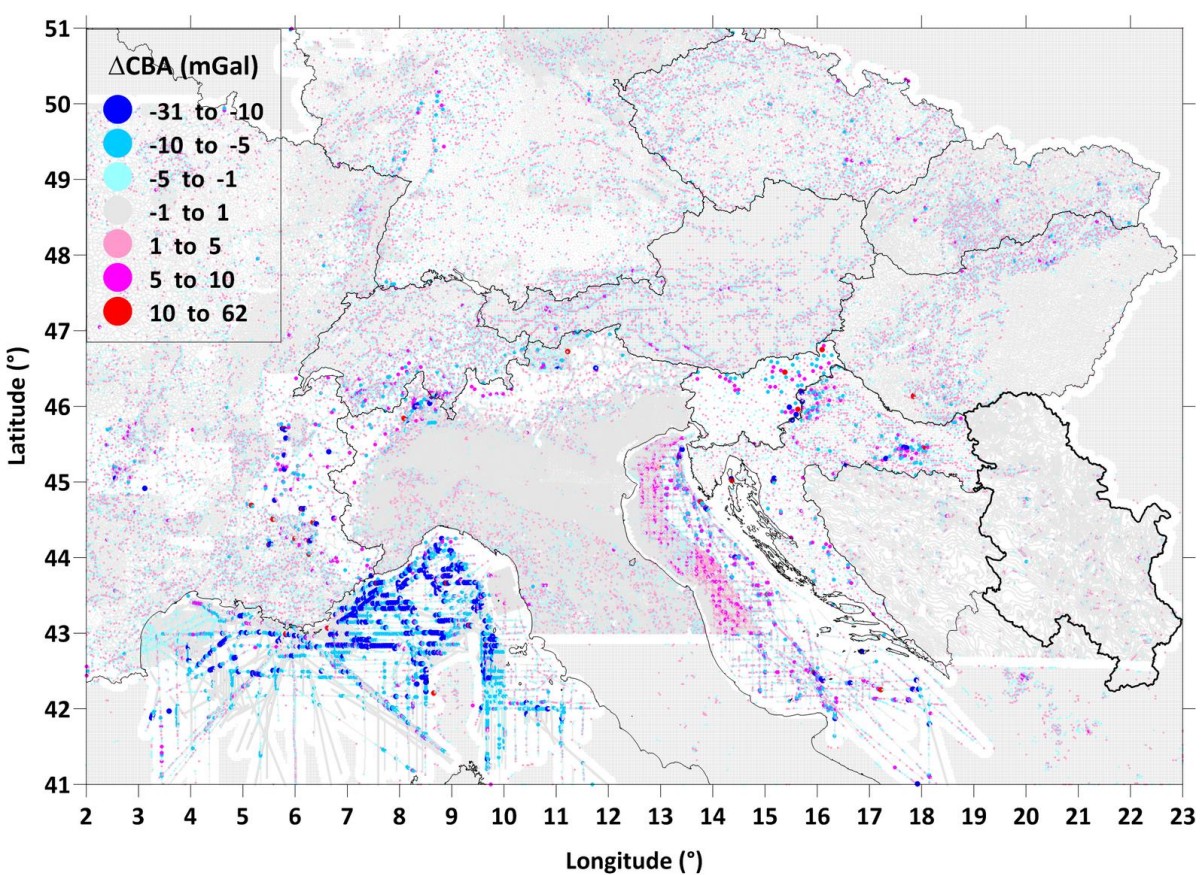

**Figure 24: Results of cross validation of the new CBA. The point sizes are proportional to the magnitude of residuals. The grey "background" represents locations with lowest residuals.**

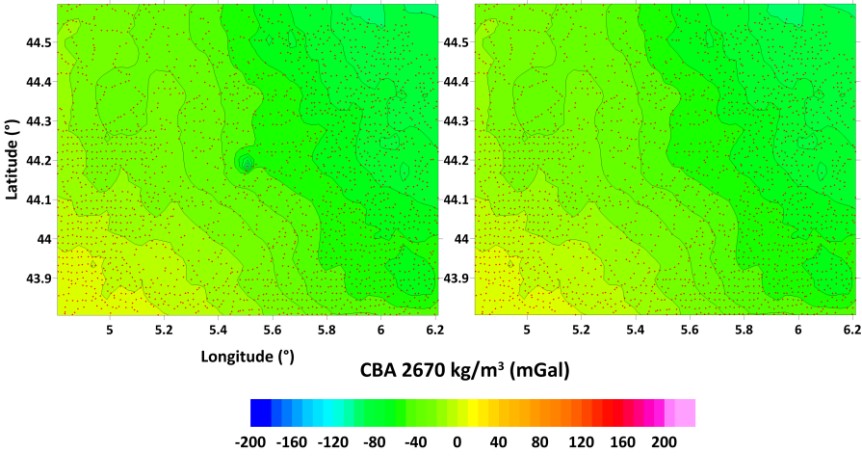


**Figure 25: Example of an extreme value of more than 60 mGal deviation in the new CBA map: initial CBA version (left) and final CBA version (right). Small red markers represent data points.**



subsections. We decided to use the exclusion criterion of points exceeding Interpolation residuals ±10 mGal. A total of 733 points were excluded (Fig. 26). Except for a few points, almost all excluded points cover marine data, which confirms the

naturally lower quality of marine data.

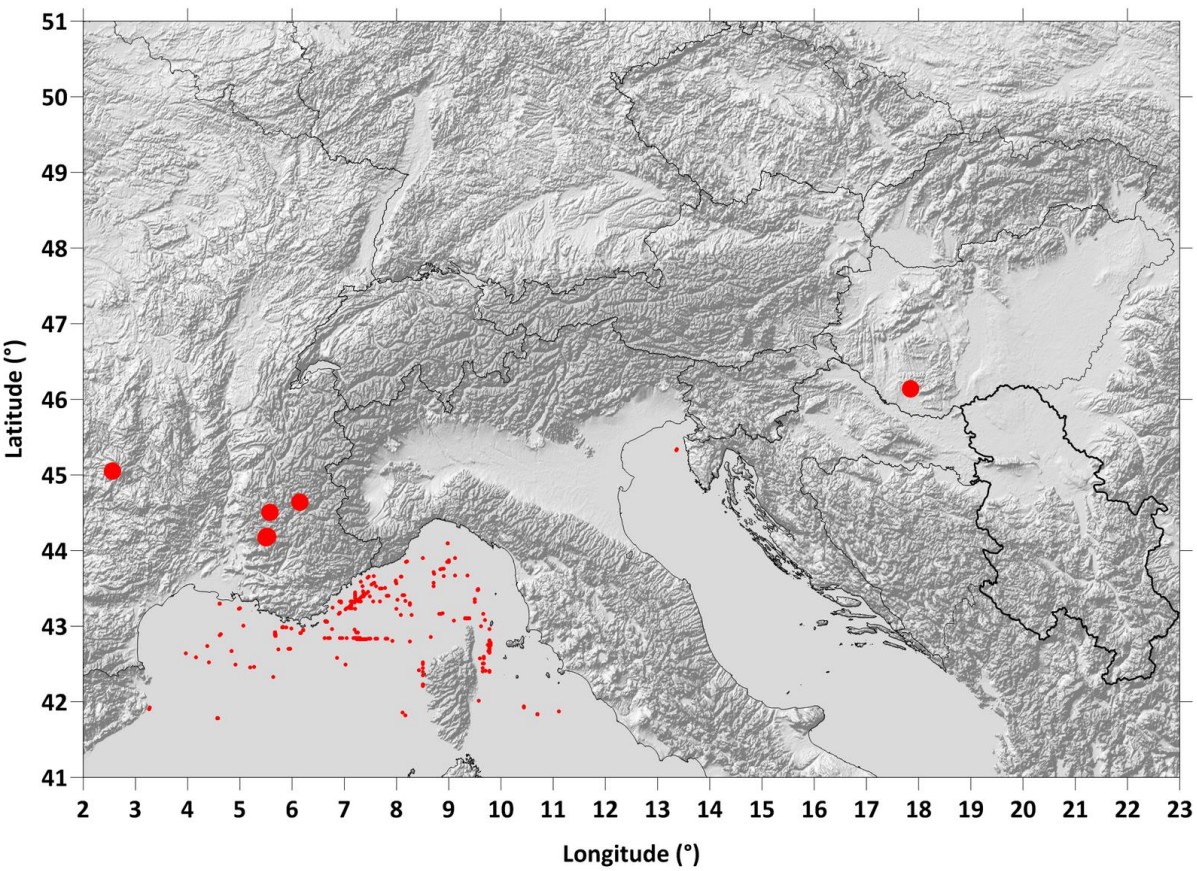

**Figure 26: Position of excluded points (733 points in total) based on interpolation residuals higher than ±10 mGal. Almost all excluded points belong to marine data, very few points lie on land (enlarged points for clarity). The shaded relief in the background shows topography to distinguish land and offshore areas from each other.**

**7 Availability of the digital data sets and criteria of use**

From the outset, the AlpArray (AA) initiative was organized in several research groups that were to contribute to the solution of very specific issues. Their main task was to organize and, where appropriate, coordinate the activities of all members within the group. Of the six AA research groups, five were concerned with the solving of seismic problems, and the sixth group had set itself the task of uniformly processing and publishing modern, homogeneous gravity anomalies of land-based gravity data.

The results of this group are here presented to the public in two grid versions. In the following, we provide readers (1) with information on the coverage, the acquisition of the data sets, and the quality of processed data and (2) their citation, long-term archiving in a data repository and DOI allocation for research data.



## 7.1 Products

At an early stage, the AAGRG considered which gravity field anomalies in an interdisciplinary work environment could

contribute to solving the principal questions posed in the AlpArray program. We hereby make the following anomaly data sets available to the community:

- Free Air anomalies (reconstructed from interpolated Bouguer anomalies according to Eq. (13)
- Complete Bouguer anomalies.
- In addition, the values of the mass/bathymetric correction will be released in a similar format to the anomalies. Their

knowledge is essential because the specification of the values for the mass correction allows an individual recompilation by the user with a different correction density. This is particularly recommended if the use of an individual density is preferable to the standard density of 2670 kg m$^{-3}$ in the area under investigation.
- Also included is the grid of ellipsoidal heights

The new gridded data sets for the Alpine gravity anomalies are published:

1345       - for the public on a grid of approx. 4 km × approx. 4 km and

      *- for the internal working groups of the AlpArray Initiative on a grid approx. 2 km × approx. 2 km.*

### Coverage and description of data tables

The area covered includes not only the core Alpine regions of the Western and Eastern Alps and the Carpathians but also parts of the Northern Apennines, the Dinarides, the Pannonian Basin and extended Alpine forelands and parts of the Adriatic Sea

and the Ligurian Sea. The lower left map corner is located at coordinates 2° E, 41° N, the upper right at coordinates 23° E, 51° N.

### Relevant specifications

### Pan-Alpine_Gravity_database_2020.dat

This file contains all results, organized into 7 columns: Lon, Lat, EH, CBA, FA, MC, BC, which respectively correspond to

Lon = Longitude (decimal degrees, ETRS89), Lat = Latitude (decimal degrees), EH = Ellipsoidal Height (m), CBA = Complete Bouguer anomaly (mGal), FA = Free-air anomaly (mGal), MC = Mass Correction (mGal), BC = Bathymetric Correction (mGal).

### Format digital grids

The five digital grid files

      "Pan-Alpine_2020_Bouguer_gravity_anomaly_grid.grd",

      "Pan-Alpine_2020_free-air_gravity_anomaly_grid.grd",

      "Pan-Alpine_2020_mass_correction_grid.grd" and





„Pan-Alpine_2020_bathymetric_correction_grid.grd"

„Pan-Alpine_2020_ellipsoidal_height_grid.grd"

are preceded by a header, followed by the array of values as described below:

- Nx  Ny                   number of longitude/latitude nodes
- Xmin  Xmax               minimum and maximum values in longitude
- Ymin  Ymax               minimum and maximum values in latitude
- Zmin  Zmax           minimum and maximum values of anomaly
- Z1  Z2  Z3  Z4           Array of anomaly values; bottom left as the origin (0,0)
                           of the coordinate system.

Table 8 provides map-relevant information.

| Map interpolation | Kriging |
|---|---|
| $\Delta\lambda$ in geographic coordinates *) number of nodes: | 0.025 990 1° 809 |
| $\Delta\varphi$ in geographic coordinates *) number of nodes | 0.017 985 6° 557 |
| Lower left corner | 2°E, 41°N |
| Upper right corner | 23°E, 51°N |
| Coordinates system | ETRS89 (ellipsoid GRS80) |
| Grid size *(for public download)* | 4 km × 4 km |
| Grid size *(for internal AlpArray download only)* | 2 km × 2 km |


**Table 8: Summary of map-relevant information. *) Note: The agreed area boundaries do not fit exactly with the proposed grid step: so, it was decided to fix the area boundaries and numbers of nodes in longitude and latitude direction. This resulted in somewhat skewed spacing values.**

**Bouguer Gravity Map**

Although it was and is the declared objective of the AAGRG to compile digital gravity data for the Alps and their adjacent

areas, a high-resolution Bouguer gravity map is also available for download in PDF format (Supplement). Besides the anomaly

in form of a "heat map" it also contains geographic information for better orientation. Fig. 27 shows the spatial distribution of all original data considered for the map compilation and all areas where GGM data have been used for filling gaps (refer to Sect. 5.2).


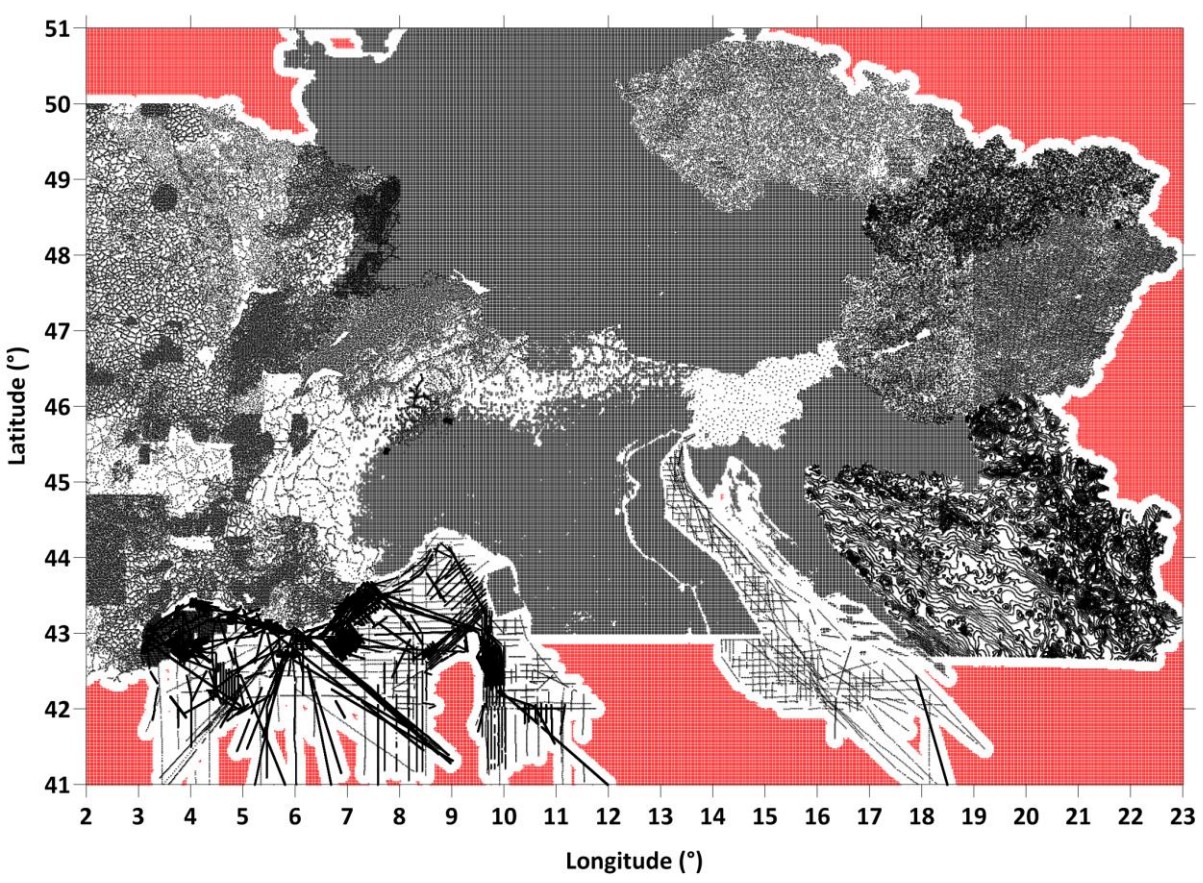

**Figure 27: Despite all efforts to achieve the greatest possible homogeneity in the data basis and processing steps, this map is intended to show that the initial data basis was different due to national requirements. First, the outer areas shown in red are supplements/fillings with GGM values (Sect. 5.2). Irregular black dots indicate the use of point data and in the offshore areas of the Ligurian Sea and the Adriatic Sea the black lines indicate the ship tracks. In the southeast of the chart, isolines have been digitized (see also Sect. 4.5).**


### 7.2 Long term archiving, and downloads

Research data like the digital gravity data base for the Alps and adjacent areas are an essential basis for scientific work. The variety of data corresponds to the diversity of different scientific concepts, knowledge interests and research methods. The long-term storage and access to the data contributes to the reproducibility and quality of scientific work and opens important possibilities for further research. The Alliance of European Science Organisations has already declared its support for the long-

term storage of open access to consideration of disciplinary regulations in the handling of research data in the "Principles for
Handling Research Data" adopted in 2010 (DFG, SNSF, etc.). The publication and storage of the pan-Alpine gravity data and
the accompanying Bouguer gravity map follows these standards. After the geophysical project completion described in the last
section the group is obliged for various reasons (e.g., AAGRG "Memorandum of Collaboration" with the participating
countries, long-term value of the data) to store the data permanently.

The AAGRG decided to publish the research data produced with GFZ Data Services (http://pmd.gfz-
potsdam.de/portal/about.html). GFZ Data Services is the cooperation partner for data publication via the specialist information
service (FID.GEO). (https://www.gfz-potsdam.de/zentrum/bibliothek-und-informationsdienste/projekte/fid-geo/). The
German Research Centre for Geosciences GFZ, the operator of GFZ Data Services, has been issuing Digital Object Identifiers
(DOI) to data sets since 2004 in accordance with the principles of the International DOI Foundation (https://www.doi.org/).
These data sets are archived and published by GFZ Data Services and cover the entire range of geoscientific activities.


In order for the gravity data to be found worldwide on the Internet, the data must be given a description that is readable by
search engines. This description is provided by **metadata**. The specific description of metadata for our data set is important
but is not part of this publication. Additionally, for other users to be able to evaluate and reuse our data, the data must be
supplemented by a verbal description in addition to the metadata, explaining the data, its processing etc. to others. This
publication meets this requirement.

### Data ownership

Data access and use is defined by the AAGRG. The copyrights and access rights are described in a license which is firmly
attached to the data and defined in which way the data may be used or not.

### Licences

The article and corresponding preprints are distributed under the Creative Commons Attribution 4.0 License. Unless otherwise
stated, associated material is distributed under the same license.

### Publication of data with a time embargo

In principle, the setting up of blocking periods is possible. In this case, the data, after they have been prepared for publication
in the GFZ system, are not open to the public during the embargo period. But there are already certain advantages, because the
publication can be found worldwide via the publicly accessible metadata of the gravimetric datasets and with the assigned DOI
the data can be cited even before the end of embargo.





## 8 Data availability

For the new data sets also a DOI was assigned. The data will be published with the DOI

https://doi.org/10.5880/fidgeo.2020.045 (Zahorec et al., 2020) when the final paper is accepted. In the meantime, the data is accessible via this temporary review link: https://dataservices.gfz-potsdam.de/panmetaworks/review/fdc35a9f6551b01b6152ee1af7b91a5a0c3de5341d067644522c192ad7f25e7f/

The data are stored permanently and available in the data repository of the German Research Centre for Geosciences GFZ. The GFZ has been publishing geoscientific research data since 2004 and guarantees technical integrity and long-term

availability.

## 9 Conclusion

The aim of this publication is to report on the activities and work of the AlpArray Gravity Research Group (AAGRG) over the last more than three years. The group´s mission was to recompile and release digital homogenized gravity data sets that are

based on terrestrial gravity measurements which are owned by the national Alpine neighbouring countries (in total more than 1 million data points). It can be used for high resolution modelling, interdisciplinary studies from continental to regional and even to local scales, as well as for joint inversion with other datasets. Bouguer and Free Air anomalies are available at a grid density of 4 km × 4 km for the public and of 2 km × 2 km for internal AlpArray use on request. The final products will also include grids for mass/bathymetric corrections of the measured gravity at each grid point. This allows the use of later

customized densities for their individual calculations of mass corrections between the physical surface and the ellipsoidal reference.

Both digital data sets are compiled according to the most modern geophysical and geodetic criteria and reference frames (both location and gravity). This includes the concept of ellipsoidal heights and implicitly includes the calculation of the geophysical indirect effect; atmospheric corrections are also considered. For the calculation of station completed Bouguer anomalies we

used the following densities: 2670 kg m$^{-3}$ for landmasses, 1030 kg m$^{-3}$ for water masses above and -1640 kg m$^{-3}$ below the ellipsoid. The mass correction radius was set to Hayford zone $O_2$ (167 km). Special emphasis was put on the numerous lakes in the study area. They partly have a considerable effect on the gravity of stations that lie at their edges (for example, the rather deep reservoirs in the Alps). In the Ligurian and the Adriatic seas, ship data of the Bureau Gravimétrique International were implemented in the digital database. Although not unproblematic, these data got the preference over satellite data offshore.

In the future, the calculation of long-distance effects of topography/bathymetry and its compensating masses (roots) are planned. Absolutely necessary is a more profound analysis of the map uncertainties. The associated research is complicated by the fact that for many of the national data sets used, no metadata are available. The reasons for this are manifold and do not lie with the group. To obtain an estimate of the error size in the present compilation, cross validations were calculated, both

for the entire grid and for the national grids. After an iterative improvement by elimination of erroneous data, a map error of about ± 5 mGal can be assumed after the third iteration. In some offshore areas the error is less than 10 mGal.

**Supplement** "High resolution_Pan-Alpine_2020_Bouguer-anomaly_map_600spi.zip"

**Authors contributions**. While a separate page could be filled with detailed author contributions, we highlight the great joint effort that resulted in the current paper, and, within that, that all authors contributed to Data curation as the most crucial step. The Conceptualization stems from the AlpArray program and included authors GH, H-JG, CB, RP, PZ, JP, MB, JE, GG, Ago, BM, JS, PS, ES, MV.

The Methodology and Formal analysis rested mostly on the shoulders of PZ, JP, RP, BM, with help from many others, at least one person per country for Investigating national datasets.

The original draft was written by H-JG, PZ, JP, MB, BM, GH, RP, SB and CB. All co-authors have reviewed and approved the final version of the manuscript.

Funding was acquired on a national basis, as described in the acknowledgements.

H-JG and GH ensured the overall Project administration and Supervision.

**Competing interests.** The authors declare that they have no conflict of interest.

**Acknowledgements**

This first compilation of gravity databases in the Alps and their surroundings has many helpers and supporters in the national administrations, state offices, the academic working groups of the participating universities, the Steering Committee of the AlpArray program and finally in the Library Services of the GFZ (Potsdam/Germany). We would like to thank them all very much for their support. The group met several times for two-day working meetings and we gratefully thank the organizers. The work of members of the AAGRG was also supported by national funding: In Germany by the German Research Foundation (DFG, INTEGRATE project, SPP 2017). In Hungary, the research has been partly funded by the National Research, Development and Innovation Office Grant. 124241. In Italy, the research has been partly funded by a MIUR Department of Excellence funding, for the Department of Mathematics and Geosciences, University of Trieste. In Switzerland, the Schweizerischer Nationalfonds (SNF, grants PP00P2_157627 and PP00P2_187199) supported the studies. In Slovakia the work was accomplished with partial support from grants by the APVV agency by means of project APVV 19-0150, APVV 16-0146 and APVV 16-0482, and the Slovak Grant Agency VEGA by means of project 2/0006/19 and 2/0100/20. The results



of this study contribute to the activities of EPOS (European Plate Observing System) and its French component RESIF (Réseau sismologique et géodésique français).

Finally, we would also like to thank Tomislav Bašić, Josip Stipčević (Croatia) and Kirsten Elger (Germany) for supporting our work.

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
