# Peer review of "The first pan-Alpine surface-gravity database, a modern"

_Earth System Science Data, 2020_

## Author Response (AR1)

**Rebuttal letter**
Manuscript: The first pan-Alpine surface-gravity database, a modern compilation that crosses frontiers

By P. Zahorec et al.

Submitted to: ESSD; Preprint **essd-2020-375**

Dear Topical Editor Christian Voigt, dear Roland Pail, dear Anonymous reviewer,

As we wrote in the discussion phase of the manuscript, we would like to thank you for the time invested in reviewing our manuscript. We are pleased that the assessment of the content and the significance for the geo-community is positive, also documented by positive public comments by Prof. Scheck-Wenderoth and Prof. Weber. In addition, I had also sent the manuscript to Prof. Rummel for an "internal" review. His comments have also been incorporated into the new manuscript version, in addition to those of the topical editor and the two reviewers. After intensive internal discussions among the 27 authors, we largely followed the suggestions of the topical editor.

**(I) With respect to the comments by the reviewers:**

Roland Pail has acknowledged in his review the work done by the international AAGRG over the last years and we believe that he has fully accepted our intention in writing the manuscript - we appreciate his review very much. He had also expressed that he agreed with a revision of the original manuscript, as I had already communicated during the discussion phase.

Regarding the anonymous review, we would like to address some aspects of his review in detail.

(1) *However, the paper is just a technical report with a documentation of all details possible with regard to the collected data but does not include any new and/or innovative topics (besides perhaps the interpretation given at the end of the paper). The paper is fine as a technical report but should not be published as a paper in a journal. My suggestion is to publish the paper in its present form as a technical report (e.g., within a series maintained by the participating institutions or online) and to re-submit to ESSD a significantly shortened and more concise version of the paper on the main topics and results.*

Our comments:
ESSD expects a precise and comprehensive, even detailed description of the data sets provided. The presented dataset is the result of national data collection of the Alpine countries for almost 100 years applying different techniques of data acquisition and processing. Therefore, detailed explanation as provide by us is very much needed.

We disagree with the suggestion to publish the manuscript as a report at one of the participating institutions. The technical description is novel and very much needed to

comprehend the details of the data sets. If the interest is only in applying the data on a regional scale, such details might not be needed, but for applications interested in the details of the data sets, that is critical information.

(2) *Especially the history and database details on a country-by-country basis (pages 4 to 15) should be condensed and go to an appendix. Also, the list of abbreviations and other detailed information could go to an appendix. Besides the abovementioned history and database details, the paper is lengthy in almost all sections with too many details, not suited for a paper (e.g. reference systems, digital elevation models, mass corrections, validation, etc.).*

Our comments:
We also disagree that our work does not contain "something new and/or innovative topics" (also refer to our comments on suggestions of the Topical Editor below) and with the recommendation that "the historical data details" should be condensed as they contain important meta-information that was/is indispensable for the overall data processing. We have asked ourselves why "reference systems", used "DEMs", explanations of "mass correction and validation" should be superfluous. We consider the evaluation of the "uncertainty" of the new CBA data as an important point.

However, we followed to a large extent these suggestions and pushed the list of abbreviations and the manuscript parts with the historical contributions into appendices. Also, parts of the remaining manuscript main body have been rewritten. Please see also our replies to the topical editor.

**(II) Regarding the suggestions of the Topical Editor, we would like to respond as follows.**

As the Topical Editor agrees on, the presented manuscript will surely attract a wide and diverse readership, already now downloads of the preview on ESSD are numerous with about 700 accesses.

(1) *Condense your manuscript and write a concise version on max 15-20 pages to keep it attractive to every reader, while considering the rules of writing a good scientific manuscript and*
(2) *Refer the interested reader to a greatly expanded appendix.*

Our comments:
While we do not consider that the length of a paper should be subject for criticism, if the content requires such details, we put substantial chapters (e.g., the history chapters and the contributions of individually involved countries, metadata description, list of abbreviations and some short technical chapter parts) in four appendices to increase appeal to the reader:

Appendix A (line 896): Abbreviations,
Appendix B (line 968): Historical remarks on alpine gravity surveys and national gravity
        databases for AAGRG Bouguer gravity compilation,
         - Historical activities &
         - National contributions.
Appendix C (line 1260): Digital elevation models in the AlpArray region and
Appendix D (line 1340): Mass correction – software and comparisons.

For the main part of the manuscript, we have actually 34 (WORD) pages, because we think that the numerous performed processing steps must be clearly preserved in the description of the main body.

(3) *For the guidance of the reader, a table of content might be helpful.*
Under the first chapter (1 Introduction) we included a sub-chapter "Publication layout" (line 98) that will help to guide readers back and forth between the main body of the publication and its appendices.

(4) *One final remark: Most of the authors should be familiar with the European Gravimetric Quasigeoid project (EGG) by the IAG: ….*

Our comments:
Concerning the actuality and novelty of the data compilation we carried out, we also would like to reply to the opinion of the reviewer 2 that this compilation would not result in anything new. On the contrary, thousands of *unpublished* (e.g., industry) data have been included in the compilation due to the contracts we have signed with the participating countries. These data will neither be shown as point data, nor described as such. By agreement we can publish the new database only on a 4 km x 4 km grid and give the 2 km x 2 km grid exclusively to members of the AlpArray consortium only after written consent of the participating institutions.

This is also the significant difference to the noteworthy work of Heiner Denker (2013) , whose work for the IAG is now mentioned in our manuscript. We mentioned H. Denker´s work in line 96 explicitly.

The database described and published here is unique in this respect also justifies the designations "first... Bouguer gravity database" etc. Furthermore, we would like to mention that neither an improvement of the quasigeoid nor of the geoid in the difficult high mountains of the Alps was in our focus. Our aim is to provide a valuable contribution for the numerous interdisciplinary projects in the AlpArray initiative and other European geo projects (see also comments by Prof. Scheck-Wenderoth and Prof. M. Weber) that supports lithosphere and mantle modelling in the Alpine-Mediterranean region.

We were working on the complete revision of the manuscript along the suggestions of Christian Voigt's summary and hope that this urgently needed new database can be published soon. Once again, on behalf of the authors team I would like to thank the reviewers and the Topical Editor for their advice on improving the original manuscript and for their time invested.

H.-J. Götze
hajo.goetze@ifg.uni-kiel.de

---

## Author Response (AR2)

**Rebuttal letter**
Manuscript: The first pan-Alpine surface-gravity database, a modern compilation that crosses frontiers

By P. Zahorec et al.

Submitted to: ESSD; Preprint **essd-2020-375**

Dear Topical Editor Christian Voigt,

Thank you for sending your technical notes and suggestion. I followed strictly your suggestions. In detail I corrected:

*1. LL. 178-179: Better remove the statement on not using SI unit but mGal instead from page 7 to a footnote of the first mentioning in the abstract.*
DONE

*2. LL. 59-61 and LL. 865-871: The paragraph on the DOI publication has to be rewritten for final publication.*
After conversation with Dr. K. Elger I replaced the corresponding text passages by:
*"The data are published with the DOI https://doi.org/10.5880/fidgeo.2020.045 (Zahorec et al., 2021) via GFZ Data Services."*

*3. L. 256, L. 1300, L. 1397: arcsec or " (instead of sec).*
DONE

*4. L. 288, L. 1117, L. 1178, L. 1250: topographic (instead of topographical) maps.*
Done

*5. L. 355: Figure 8: histogram not readable (to be enlarged).*
DONE

On behalf of all co-authors I was working on the final revision of the manuscript along all suggestions of TE Christian Voigt and hope that this new database can be published now.

Once again, on behalf of the authors team I would like to thank our Topical Editor for his advice on improving the original manuscript and for his time invested.

H.-J. Götze
hajo.goetze@ifg.uni-kiel.de